# Multicolor lifetime imaging and its application to HIV-1 uptake

Tobias Starling[1], Irene Carlon-Andres [1], Maro Iliopoulou[2,5], Benedikt Kraemer[3], Maria Loidolt-Krueger[3], David J. Williamson [1] & Sergi Padilla-Parra [1,2,4] ✉

Simultaneous imaging of nine fluorescent proteins is demonstrated in a single acquisition using fluorescence lifetime imaging microscopy combined with pulsed interleaved excitation of three laser lines. Multicolor imaging employing genetically encodable fluorescent proteins permits spatio-temporal live cell imaging of multiple cues. Here, we show that multicolor lifetime imaging allows visualization of quadruple labelled human immuno-deficiency viruses on host cells that in turn are also labelled with genetically encodable fluorescent proteins. This strategy permits to simultaneously visualize different sub-cellular organelles (mitochondria, cytoskeleton, and nucleus) during the process of virus entry with the potential of imaging up to nine different spectral channels in living cells.

Genetically encodable fluorophores, such as the green fluorescent protein (GFP) and its derivatives enabled revolutionary improvements in light microscopy[1]. Over the past 25 years protein engineering has been extensively employed to enhance and modify the properties of many FPs[2]. Methods such as directed evolution[3], structure-based mutagenesis[4], or recently structure-based rational design[5] have allowed to reach a colorful FP palette of great variety in terms of both biochemical and spectral properties. One of the most important aspects of FP engineering has been to achieve bright FPs. The lifetime, however, is a unique spectral property that is characteristic of every fluorophore and is less studied when designing a new FP. It is a measure of the time taken for a molecule in the excited state (after absorbing a photon) to return to the ground state (and emitting a photon). It is often referred as the spectral fingerprint of fluorophores[6]. Fluorescence Lifetime imaging microscopy (FLIM) is a technique independent of the optical path or the fluorophore concentration. Traditionally, it has been employed to reliably quantify protein-protein interactions through Förster resonance energy transfer (FRET) detection[7–9]. Although developed a few decades ago, FLIM has only recently gathered attention thanks to the development of fast electronics, artificial intelligence, and the decrease of the deadtime of digital detectors capable of photon counting. In practice, this means faster acquisitions and a better temporal resolution. In addition to

these technological advances efficient and simplified ways of analyzing FLIM data have also brought FLIM to the public[10] such as the pattern matching approach[11], phasor plot[12,13], the average lifetime[14] or other simplified global analysis[15]. When considering FP engineering only a few labs have employed the lifetime as a crucial property[16]. Therefore, when consulting the literature regarding lifetime values for FPs expressed in live cells few data are available. Not always a bright FP presents a lifetime decay which could be modeled with a desirable single exponential fit[17]. This feature is not only important to simplify data interpretation it also impacts FP photostability[18]. There is currently no comprehensive list of FP lifetimes under different experimental conditions, especially when expressed in live cells at 37 °C, as the lifetime depends on temperature. Here, we have focused on the use of FLIM to characterize the most appropriate combination of existing FPs for live cell multicolor imaging. We also show that very similar FPs in terms of spectral emission can be separated based on their lifetime information employing different analytical approaches: the phasor plot, the average lifetime and fitting approaches.

Current multicolor FLIM approaches are based on combining spectral imaging with lifetime measurements[11,19]. Here, we combine the use of many FPs fully available in public repositories with commercially available technology and lifetime data to demonstrate that the right combination of FPs' lifetime, spectral properties, and multichannel

[1]Department of Infectious Diseases, King's College London, Faculty of Life Sciences & Medicine, London, UK. [2]Division of Structural Biology, Wellcome Centre for Human Genetics, University of Oxford, Oxford, UK. [3]PicoQuant GmbH, Rudower Chaussee 29 (IGZ), 12489 Berlin, Germany. [4]Randall Division of Cell and Molecular Biophysics, King's College London, London, UK. [5]Present address: Randall Division of Cell and Molecular Biophysics and Department of Physics, King's College London, London, UK. ✉e-mail: sergio.padilla_parra@kcl.ac.uk

detection is sufficient to achieve 9 color imaging. Importantly this approach is applied to living cells exposed to quadruple labeled HIV-1 viruses. Comparative analysis of different FLIM methods showing their strengths and weaknesses is carried out when quadruple labeled viruses are exposed to live cells. This combined approach allows the dissection of different steps of HIV-1 entry by also labeling key organelles of the host cell, such as the cytoskeleton (via LifeAct-mScarlet), the mitochondria (via Mito-LSSmKate2) or the nucleus (via laminB-mWasabi). We show that multicolor PIE FLIM combined with pattern matching analysis can efficiently separate up to nine species when considering live cells and seven when examining quadruple labeled HIV-1 viruses exposed to live cells.

## Results

### Selecting fluorescent proteins for multicolor FLIM

The principle of confocal FLIM is described in Fig. 1a. A broad palette of FPs with different spectral properties are excited with three alternatingly pulsed lasers, also termed pulsed interleaved excitation, PIE[20], tuned to 440 nm, 485 nm and 594 nm respectively. From a panel of 30 available FPs (Supplementary Table 1), we individually expressed and imaged each one of them in live cells. From each of the FPs' lifetime information we identified 21 pairwise to hexawise combinations in three spectral channels suitable for multiplexing up to nine colors in one acquisition (Fig. 1a).

For the specific choice of FPs, the relative brightness of each fluorophore also needs to be considered[21]. Simultaneous excitation with the same laser power for two fluorophores with similar absorption spectra but big differences in brightness will result in the prevalence of the one with higher brightness for their specific spectral channel (Fig. 1b) regardless their differences in lifetime. Therefore, it is important that FPs, in addition to having different lifetime, also have similar brightness. We have also evaluated the minimal brightness differences between fluorophores to justify our final choice of lifetime-separable FP combinations (Fig. 1b). The lifetime of each FP considered was analyzed both when expressed individually in live cells (Supplementary Fig. 1) and when expressed together by pairs for each color channel (Supplementary Fig. 2). We also analyzed how different FLIM analysis approaches affect the overall lifetime calculation for each FLIM measure on live cells (Supplementary Fig. 1).

### Pixel-by-pixel unmixing of spectrally similar fluorescent proteins

To validate our non-fitting analytical unmixing approach, two FPs with similar emission spectra and high relative brightness were chosen as a benchmark example: mTFP1[17] and mTurquoise2[22]. Both proteins have been previously characterized as having single exponential lifetimes and therefore their separation within the same pixel should be possible using a double exponential model (Fig. 2a). Employing a simplified FLIM configuration with 440 nm and 485 nm alternating pulsed lasers and one photon counting detector we were able to use our non-fitting unmixing approach to separate both fluorophores co-expressed in the cytosol of live cells employing our non-fitting unmixing approach (Fig. 2b). Even if Forster Resonance Energy Transfer (FRET) could occur between mTFP1 and mTurquoise2, this phenomenon could not be quantified as the settings to recover FRET by FLIM would necessitate the recovery of photons from the donor alone. This is not possible given the emission overlap of both blue FPs and the optical path and optical elements employed (see material and methods). Three different types of analysis were performed on the same cells: fitting approaches employing single and double exponentials deconvolved with the instrument response function (IRF) (Fig. 2a), the average lifetime approach (Fig. 2b) and the phasor plot (Fig. 2c). In all cases the same results were obtained for the three ways of expressing both FP's in live cells. For mTFP1-mTQ2 tandem ($n = 10$), the bi-exponential fitting approach yielded a proportion of mTFP1 at 53% ± 3, the average lifetime gave 49% ± 5 and the Phasor Plot found 47% ± 3. For the bicistronic mTFP1-T2A-mTQ2 condition ($n = 10$) the fitting approach gave 53% ± 3, the average lifetime 57% ± 6, and the Phasor Plot 57% ± 8. Finally, for the condition where mTFP1 and mTQ2 were co-expressed ($n = 10$) the fitting approach gave 53% ± 3, the average lifetime 47.8% ± 7, and the Phasor Plot 46.5% ± 9. Other FP couples (GFP + mWasabi; LSSmOrange + LSSmKate2 and mCherry + mKate2) were also tested applying both the average lifetime and phasor plot approach (Supplementary Fig. 2).

To further validate our non-fitting approach based on the average lifetime to unmix a two species system with different lifetimes we performed a series of analyses employing a surrogate system of two ideal solutions mixed with known proportions of two fluorophores with know lifetimes (Supplementary Fig. 3). Interestingly, we show that the signal to noise of the acquisition is more relevant as compared to small or big increases in lifetime of the two species under investigation.

### Defining the proportions of HIV-1 receptors when engaged in the virological synapse

HIV-1 can transmit directly from an infected (effector) cell to a target cell through a supramolecular structure between both cells comprised of viral Env on the effector cell bound to receptors and co-receptors on the target cell. Normally the infected and target cells are T cells, but these structures can also be seen in effector cells and other sub-types such as dendritic cells and macrophages[23] susceptible to HIV infection. The envelope glycoprotein of HIV-1 (Env) is required for the virological synapse (VS) formation even if their assembly mode with CD4 and coreceptors (CCR5 or CXCR4) is not fully understood. To show the power of our unmixing protocol we formed virological synapses labeled with two blue fluorescent proteins and examined their distribution within the VS. To achieve this we produced CD4-mTFP1 and CXCR4-mTurquoise2 fusion proteins and co-expressed them in live cells (as the "target" cells). (Supplementary Fig. 4). We also co-expressed HIV-1 Env (HXB2, an X4-tropic strain) and Gag-mCherry in TZM-bl cells (as the "infected" cells). We then combined these two cell populations together and acquired FLIM images where an accumulation of Gag-mCherry between two cells indicated a potential VS (Fig. 3). From the lifetime data, we calculated the proportions of CD4-mTFP1 and coreceptor CXCR4-mTurquoise inside and outside the HIV-1 virological synapse[24] (Fig. 3d and e respectively).

The proportion of CD4-mTFP1 right at the VS was 60% ± 15 ($n = 10$) (Fig. 3d left panel, double exponential fit), 64% ± 10 ($n = 10$) (Fig. 3d middle panel, phasor plot) and 53% ± 20 ($n = 10$) (Fig. 3d right panel, average lifetime approach). In contrast with the average outside CD4-mTFP1 proportion outside the VS that turned out to be 34% +/− 15 ($n = 10$) (Fig. 3e left panel, double exponential fit), 40% ± 8 ($n = 10$) (Fig. 3d middle panel, phasor plot) and 40% ± 20 ($n = 10$) (Fig. 3d right panel, average lifetime approach). These results suggest that CD4 is the main receptor interacting with Env right at the VS (determined by Gag-mCherry recruitment to the cell-cell interface), in agreement with the stoichiometry found for the pre-fusion reaction in single virus infection[25], where two HXB2 Env was engaged with eight CD4 receptors and a CCR5 dimer.

### Simultaneous imaging of nine fluorescent proteins in a single FLIM acquisition

The challenge to unmix a complex system of up to nine FPs in one single shot was taken considering our choice of FPs (Supplementary Fig. 1) and a commercial turnkey system. In order to analyze the data we employed three approaches: the pattern matching approach[11] (Supplementary Figs. 5−8 and Fig. 4) which fully employs both the spectral and lifetime information, the phasor plot and the average lifetime. Three sets of HEK293T cells were each transfected with a combination of 2 to 3 unique DNA plasmids containing the FPs

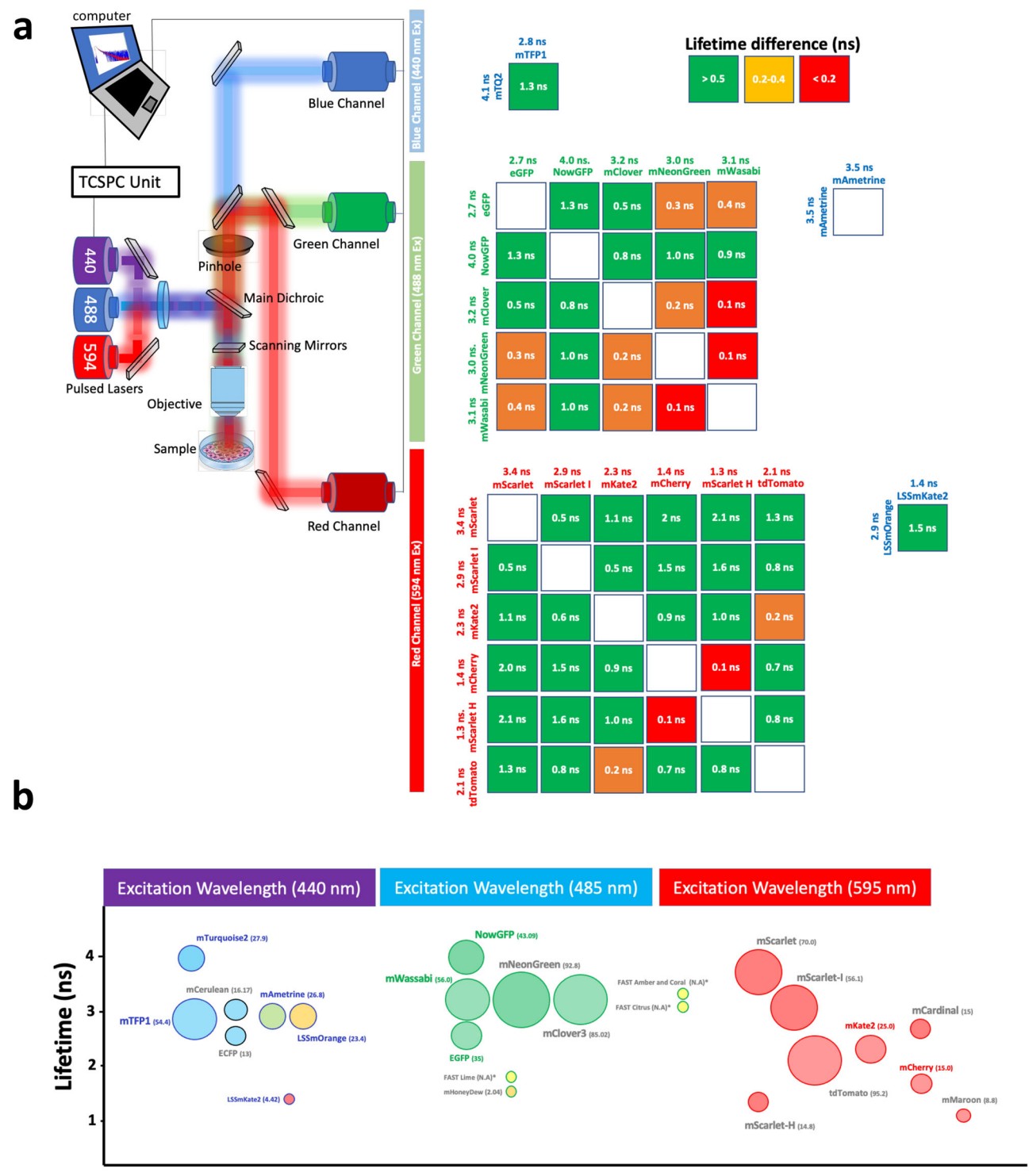

**Fig. 1 | FLIM strategy for multi-color imaging of fluorescent proteins.** A commercial time domain time correlated single photon counting (TCSPC) equipped with three pulsed lasers and three detectors (**a**) was employed. The right panels for each channel show the increase in lifetimes between two fluorophores to be resolved (in this case of the blue channel mTFP1 – mTQ2). When the increase is >0.4 nanoseconds (ns), we consider this difference to be good enough to separate two fluorophores based on their respective lifetimes and we assign a green color. When the lifetimes increase gets closer, we denote these colors orange (0.2–0.4 ns) and red (<0.2 ns) respectively. We have calculated this lifetime increase for each channel (blue, green, and red). We employed two fluorescent proteins per channel with the right spectral (in nanometer, nm), lifetime and brightness (**b**) properties to achieve eight multicolor imaging. The main two criteria employed for each channel was the difference in lifetime for similar emission spectral channels (**a**) and similar brightness (the higher the brightness the bigger the circle's diameter) The color of the circle indicates the predominant emission for each fluorescent protein.

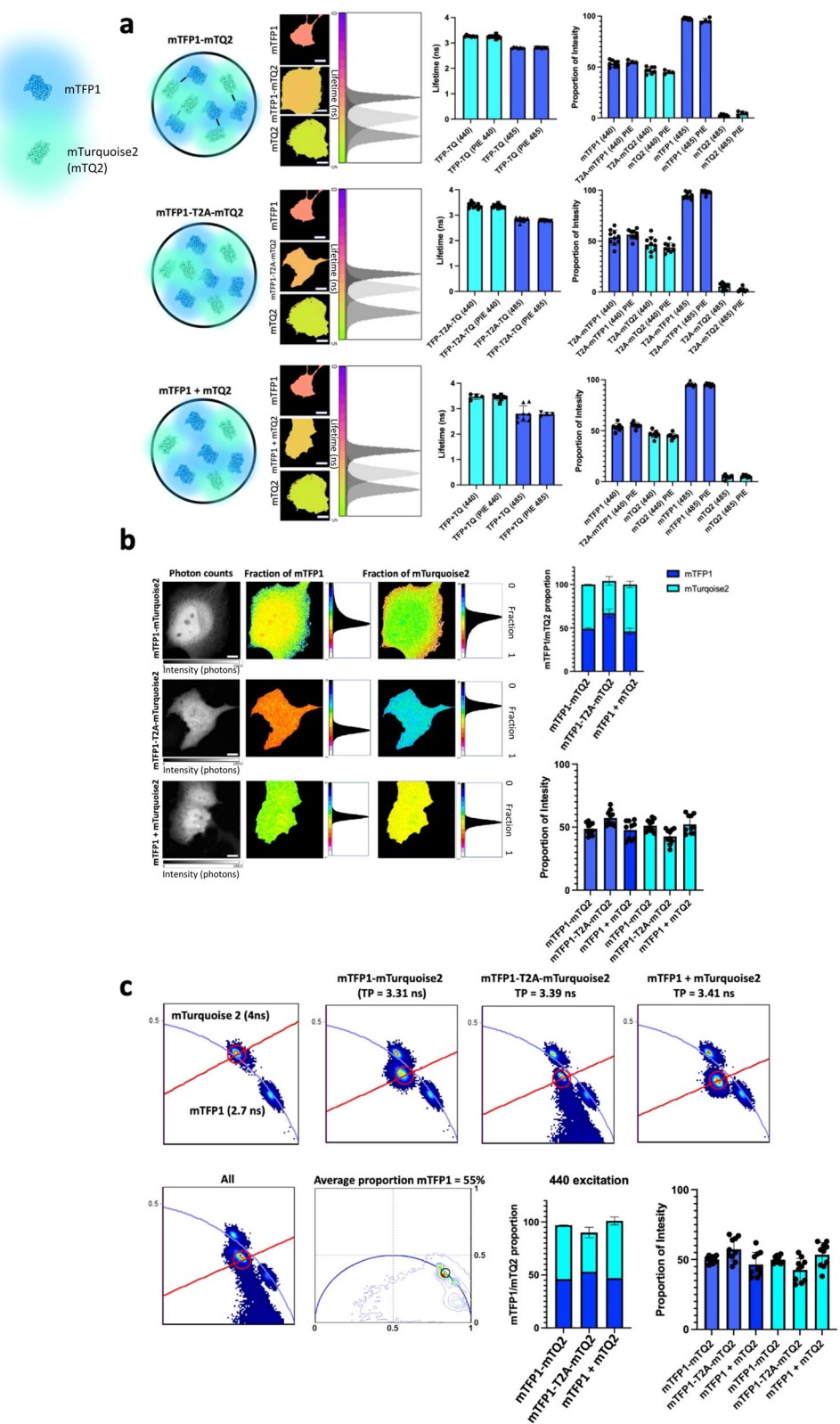

characterized previously (Supplementary Figs. 1 and 2). These cells were mixed, and a single acquisition was enough to then unmix up to nine different FPs within the observation field. By using large Stokes shift FPs (LSS-FPs) we were able to employ the red detector twice (one for the 440 nm line and another for the 594 nm) and increase the number of FPs to be multiplexed. The combination of PIE and LSS fluorescent proteins allowed us to increase the number of

fluorophores to be imaged with the red detector: one FP couple for laser 440 nm (LSSmOrange + LSSmKate2) and one FP couple for laser 594 nm (mKate2 + mCherry). The main caveat of this approach is that most of these LSS-FPs exhibit low brightness (Fig. 1b) and therefore their use is limited by the sensitivity of detectors and laser power employed. Despite these limitations we could detect and unmix both Mito-LSS-mKate2 and LSSmOrange[26] expressed in the cytosol

**Fig. 2 | Unmixing of two fluorescent proteins in the same pixel. a** Live cells (middle panels) expressing tandem mTFP1-linker-mTurquoise2 (mTFP1-mTQ2), mTFP1-T2A-mTQ2 which expresses equimolar amounts of both fluorescent proteins separately and co-expression of mTFP1 and mTQ2 with transient transfection were tested to unmix both populations of plasmids. The lifetime histograms in nanoseconds (ns) corresponding to each fluorescent protein expressed separately (dark gray) and together (light gray) for each case is also shown, the lifetime histograms present a color bar that goes from 0 (violet) to 5 ns (green). Scale bar 5 μm. Bar charts (right panels) plotting the lifetimes of mTFP1 (dark blue, 2.8 ns) and mTQ2 (cyan, 4.12 ns) when expressed alone were employed as a reference as explained in methods. The percentage obtained for the three cases depicted were $53 \pm 4\%$, $n = 8$ (tandem mTFP1-mTQ2); $53 \pm 7$, $n = 10$ (mTFP1-T2A-mTQ2) and $53.5 \pm 4\%$, $n = 8$ (mTFP1 + mTQ2) where n is the number of cells examined over 3 independent experiments; respectively. **b** The same cells shown in (a) are here analyzed with the average lifetime non-fitting approach. Micrographs show cells expressing mTFP1-mTQ2 (first row), mTFP1-T2A-mTQ2 (second row) and mTFP1 +

mTQ2 third row. The intensity bars in the micrographs denote: number of photons (first column). The pixel-by-pixel fraction of the corresponding fluorophore and corresponding histograms are also shown for comparison and validation with the fitting approach (middle and right columns). Scale bar 2 μm. The average proportions where $48.5 \pm 5\%$, $n = 10$ (tandem mTFP1-mTQ2); $57.3 \pm 6$, $n = 10$ (mTFP1-T2A-mTQ2) and $47.8 \pm 7\%$, $n = 10$ (mTFP1 + mTQ2) where n is the number of cells examined over 3 independent experiments. **c** The Phasor Plot was employed with the same cells and similar results were obtained, both in terms of lifetime and proportions, for mTFP1 and mTQ2. Both phasor signatures for mTFP1 and mTQ2 fall in the universal circle indicating that these two fluorescent proteins behave as single exponentials. The bar charts show the proportions for mTFP1 where $47 \pm 3\%$, $n = 10$ (tandem mTFP1-mTQ2); $57.4 \pm 8$, $n = 10$ (mTFP1-T2A-mTQ2) and $46.5 \pm 9\%$, $n = 10$ (mTFP1 + mTQ2) where n is the number of cells examined over 3 independent experiments. Data are presented as mean values +/- SD" as appropriate. Source data are provided as a Source Data file.

(Supplementary Fig. 2) as well as mAmetrine[27] which is also a LSS-FP, but with the maximum emission in the green channel (Supplementary Table 1 and 2).

When applying the three FLIM analyses, one can see that similar results were obtained. The patterns for each filtered image coming from all channels are very similar, highlighting the same features for the three methods (Fig. 4). All nine reference patterns are used simultaneously for linear unmixing of the multicolor image. The algorithm calculates the intensity of each pattern in every image pixel. A non-negativity constraint ensures that all intensities of the resulting images are positive values. The result is a black / white image for every pattern, as shown in Fig. 4a. An additional image in Fig. 4a shows the residuals of the least squares fit, which can indicate if the employed patterns were sufficient to describe the data in every image region. For example, low signal levels at the edge of cells or autofluorescence of certain structures could hinder the unmixing, leading to higher residuals. Pattern matching does not make any assumptions about the shape of the decay, unlike the average lifetime approach described above.

To evaluate the performance of the pattern matching approach the residual crosstalk between the unmixed images was calculated as described in Winter et al., 2017[28]. The results are summarized in Supplementary table 3. The residual crosstalk between most channels is below 15%, which shows a good performance of the algorithm. One should keep in mind that the relative brightness values of the different labels influence the crosstalk result. For example, signal bleed through of a very bright channel into a dim channel will have a relatively big impact on the resulting image of the dim channel, but the other way around it will be hardly noticeable. For illustration, the unmixed contributions of eGFP, mTFP1 and mWasabi are shown in Supplementary Fig. 8c–e. All of these FPs have very similar excitation and emission spectra and lifetime decays, as shown in Supplementary Fig. 8a, b, which would make it impossible to discriminate them based on either lifetime or spectra alone. The bleedthrough of mWasabi is notable in the eGFP and mTFP1 channels, which corresponds to crosstalk values of 0,8 % and 5,3 %, respectively. The bleedthrough of mTFP1 into the eGFP channel is also visible, which corresponds to a crosstalk value of 2,4 %. The bleedthrough of eGFP into the mWasabi channel contributes 1,8 % to the total image intensity. Some channels have very high crosstalk values of over 40 %. These belong to structures which spatially overlap in the image, such as AlphaActinin-mCherry and cytosolic LSSmOrange, or Mito-LSS-mKate2 and cytosolic mKate2. The reason is that the algorithm cannot discriminate between spectral bleedthrough and true colocalization.

Since pattern matching uses the spectral combination coming from the three detectors and PIE as a spectral fingerprint of each florescent protein, it takes advantage of the bleedthrough as a spectral reference that helps discriminating proteins. Therefore, pattern

matching turned out to perform better than the phasor and average lifetime approaches for this particular combination of FPs. This is only true because we did not take into consideration spectral FLIM information for phasor or average lifetime approaches. But, importantly, in this case we were not employing a spectral FLIM detector.

## A quadruple labeled HIV-1 virus

We produced HIV-1 viruses strategically labeled in four different viral compartments. The HIV-1 lipid bilayer (envelope) was labeled with a far-red lipophilic dye (DiD); we also labeled the Gag polyprotein complex with mCherry (termed iCherry as upon maturation the viral protease cleaves Gag releasing mCherry into the matrix)[29]. Finally, both the viral protein Vpr and the capsid (CA) were also labeled with mTurquoise2 and eGFP respectively (Fig. 5a). The impact of these markers on functionality was investigated and results showed that the quadruple labeled viruses were infectious (Fig. 5d). Of note, there was a big decrease in virus functionality for the quadruple labeled HIV-1 viruses as compared to the wild type alone. The lifetimes obtained for all label strategies (single, double and quadruple labeled particles were similar in all cases; with mTQ2-Vpr = 3.2 ns +/− 0.1; Ca-GFP = 2.1 +/ −0.1 ns; Gag-mCherry = 1.2 +/−0.1 ns and DiD = 1.7 +/−0.1 ns). We speculate that the viral environment must have a big impact in mTurquoise2 and Capsid-GFP since the lifetime of mTQ2 when expressed alone was 4.1 ns, as observed by others[30]. Regardless of the cause for this lifetime diminution for Vpr-mTurquoise as compared with mTurquoise2 expressed alone in live cells (Supplementary Table 1), we could identify and assign characteristic lifetimes for each of the four labels and therefore unmix them in the proper channel (Fig. 5c).

## Quadruple labeled HIV-1 entry

To evaluate the usefulness of the quadruple labeled HIV-1 pseudotyped with vesicular stomatitis virus spike (VSV-G) we exposed the viruses to U2OS cells at two time points ($t = 0$ min and $t = 90$ min), we fixed the cells and imaged up to 9 confocal slices (z resolution of 0.8 μm) to assess virus localization in three dimensions. We performed three-dimensional confocal microscopy (3D) combined with FLIM (Fig. 6). Only at $t = 90$ min discrete sub-viral particles (mostly Vpr-mTQ2 and capsid-eGFP) were observed in the peripheral nuclear area; in some cases, capsid-eGFP was seen within the nucleus (Fig. 6a and Supplementary Fig. 9). It is out of the scope of this article to evaluate whether the capsids where intact or uncoating commenced in the cytosol. However, these particles where only found at $t = 90$ min suggesting that most likely fusion underwent previously and therefore these where productive fusogenic events. To validate this hypothesis, we run a particle detection algorithm to recover the amount of Gag-mCherry per viral particle at both time points (Fig. 6b). Clearly, both the intensity and the number of viral particles diminished over time

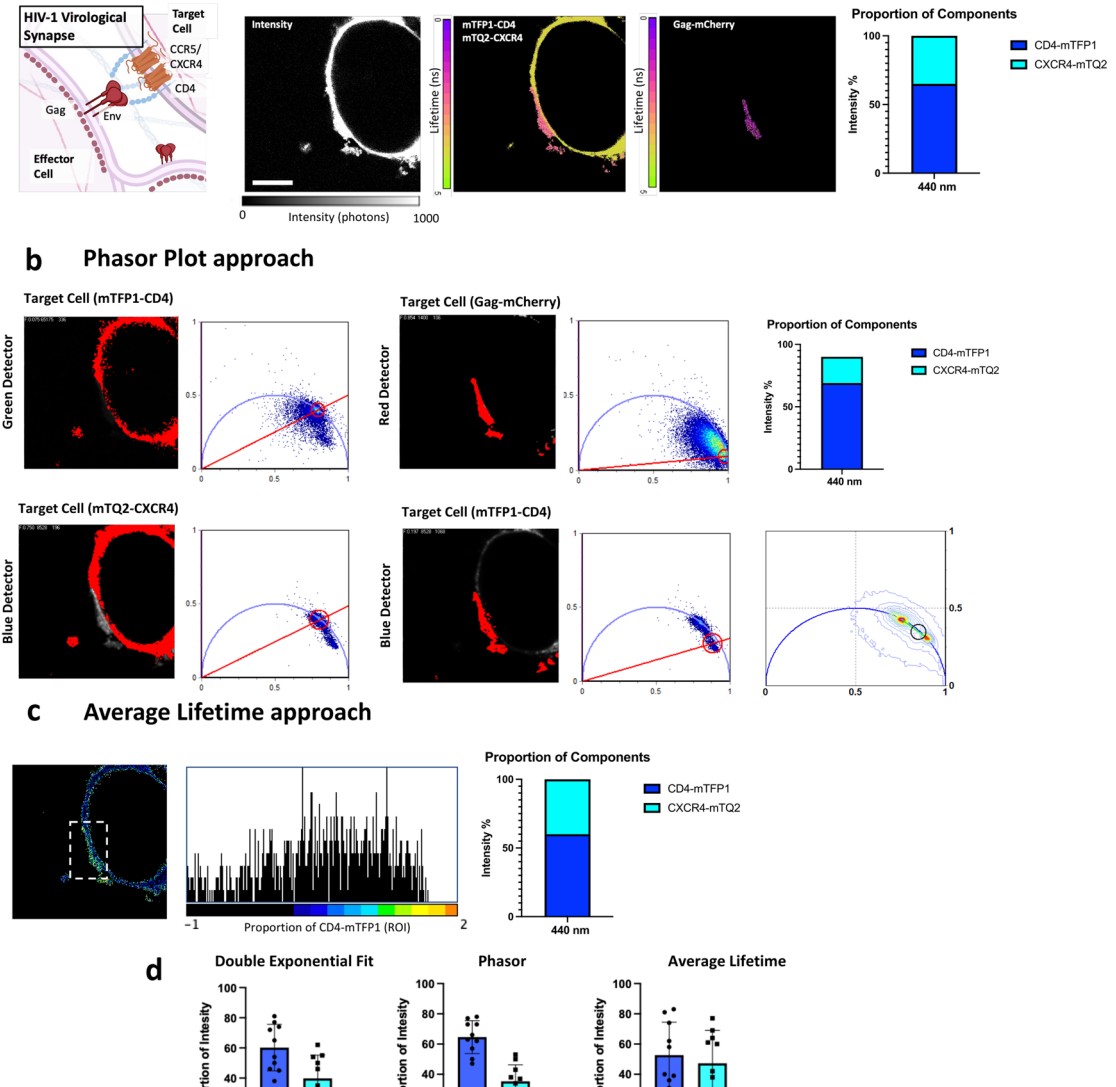

(from 298 particles localized mainly at the periphery of the U2OS cells, to 191 particles that were seen mainly in the peripheral nuclear area). The percentage of quadruple labeled particles also diminished over time for a total of 20 cells analyzed dropping from 0.65 at $t = 0$ min to 0.20 at $t = 90$ min. Willing to test whether the lifetimes of Gag-mCherry per particle were also affected when comparing both time points, we recovered the average lifetimes per virus in all cases (Fig. 6c). The

average lifetime for the viral particles at $t = 90$ min turned out to be shorter ($0.8 \pm 0.3$ ns, $n = 191$) as compared to $t = 0$ min ($0.9 \pm 0.4$ ns, $n = 298$). In both cases a $t$-test statistical analysis was performed with $p = 0.0129$, degrees of freedom = two tailed, and confidence intervals = $-0.1039$ to $-0.01239$. When plotting the frequency of lifetimes per viral particle one could see that the particles with longer lifetimes seen at $t = 0$ (lifetimes around 1.1 ns, red bars Fig. 6c, right panel)

**Fig. 3 | Proportions of CD4 and CXCR4 receptors at the virological synapse.** A biological example shows live cells engaged co-expressing CD4-mTFP1 and CXCR4-mTQ2 (target cells) and HXB2 Env and Gag-mCherry (effector cells), this experiment was repeated 10 times. **a** Intensity (left panel) and FLIM images (middle and right micrographs, in nanoseconds, ns) for both the target cell (top second panel from the left) and the infected cell (top right panel) are shown. Scale bar 5 μm. Bar diagram is shown with quantification of the proportion of components colocalizing with Gag-mCherry in the virological synapse (right panel): 65%. **b** The phasor plot approach for FLIM was also applied for the same example showing similar results as in (**a**). The cell expressing mTFP1-CD4 is shown with red pixels highlighted which correspond to the S and G coordinates of the phasor plot (right panel). The top right panels show the infected cell expressing Gag-mCherry, which demilits the region of the virological synapse with red pixels highlighted with the corresponding phasor S-G coordinates (right panel). Bottom panels show the mTQ2-CXCR4 (target cell) and corresponding phasor. The bottom right panel highlights the pixels that overlap with the virological synapse, indicating a proportion of 68% (right top panel). The phasor calculator (right bottom panel) shows the mTQ2 and mTFP1 S-G coordinates and a black circle designating the point where the VS sits. **c** The average lifetime approach is shown pixel-by-pixel for the same example as in (**a**). The dotted square shows the VS and the corresponding histogram for the fraction of CD4-mTFP1 shows an average of 59% (middle panel). This result is also ploted in a bar diagram (right panel). **d** The average proportions for CD4-mTFP1/CXCR4-mTQ2 inside VS were $60.2 \pm 15\%$, $n = 10$ (double exponential fit), $64.6 \pm 10\%$, $n = 10$ (phasor) and $53 \pm 20\%$, $n = 10$ (average lifetime). **e** The average proportions for CD4-mTFP1/CXCR4-mTQ2 outside VS were $34.9 \pm 15\%$, $n = 10$ (double exponential fit), $40 \pm 1\%$, $n = 10$ (phasor) and $40.7 \pm 20\%$, $n = 10$ (average lifetime). The number of cells for both (**d**) and (**e**) was 10 examined over 3 independent experiments for each condition. Data are presented as mean values +/- SD" as appropriate. Source data are provided as a Source Data file.

disappeared at later times (black bars same panel). This result suggests that the Gag-mCherry lifetime associated to the quadruple labeled HIV-1 virions could be a good readout to predict viral fusion as the majority of particles that lost Gag-mCherry (which can be employed as a surrogate for viral fusion[31]) had longer lifetimes of around 1.1 ns.

### Single particle detection of quadruple labeled HIV-1 virus in host cells expressing different markers for mitochondria, cytoskeleton and nucleus

We employed the quadruple labeled HIV-1 viruses to perform simultaneous seven color imaging in live cells and show the usefulness of the approach in a biological system. TZM-bl reporter cells were co-transfected with LifeAct-mScarlet, Mito-LSSmKate2 and LaminB-mWasabi (Supplementary Table 2). Observe that on this occasion we have added a fourth laser (635 nm) to excite the far-red dye DiD keeping the overall number of digital detectors to three. Snap shots were taken right after exposing the quadruple labeled HIV-1 viruses to the triple labeled TZM-bl cells ($t = 5$ min). The quantification of quadruple labeled viruses at 5 min showed quadruple labeled viruses engaged with TZM-bl cells (Fig. 7). We unmixed the 7 colors of this multicolor FLIM acquisition employing the three FLIM analyses introduced in Fig. 4: pattern matching, average lifetime and the phasor plot. These approaches allowed to recover the individual channels for each fluorescent protein and unveiled the seven colors individually (Fig. 7). The viral markers were recovered (Vpr-mTQ2, Capsid-EGFP, Gag-mCherry and the lipophilic marker DiD) together with the labels for the cytoskeleton (LifeAct-mScarlet), the nuclear staining (LaminB-mWasabi) and the mitochondria (Mito-LSS-mKate2). As in Fig. 4 one can observe that the three FLIM approaches can produce similar results in most of the seven channels. Indeed, both the phasor plot and the average lifetime highlight the same structures for, LaminB-mWasabi, Mito-LSSmkate2, Lifeact-mScarlet and DiD; note that the field of view analyzed by the phasor plot is smaller due to software analysis employed (see material and methods). For some of the channels, however, like Vpr-mTQ2, Gag-mCherry or Capsid-EGFP pattern matching highlights different patterns. We attribute these differences to the higher accuracy of pattern matching since it considers crosstalk and similar spectral components coming from the FPs studied. Even if both the average lifetime and the phasor plot can discriminate between two different components with fairly similar lifetimes the impact of spectral bleedthrough is not taken into account and only pattern matching was able to resolve the viral and cellular structures as it does take into consideration the particular signature of every FP in each of the channels evaluated.

### Discussion

Here we have shown that multicolor FLIM permits to image different groups of live cells expressing up to nine complementary fluorescent proteins on one side (Fig. 4) and quadruple labeled HIV-1 viruses exposed to triple-labeled reporter cells on the other (Fig. 7). We also show that FLIM with 4 laser PIE is possible (Fig. 7) introducing the lipophilic dye DiD excited with a 635 nm laser. The simplicity of the optical set up utilized and the limited number of lasers and detectors used (3–4 lasers and up to three different detectors) makes this FLIM multicolor approach very powerful and easy to apply also on commercially available microscopes. Importantly, other FLIM methods for multicolor imaging exploit the combination of lifetime imaging with spectral information[19].

First, we have introduced a non-fitting approach based on the average lifetime capable of unmixing a two species system of two fluorophores with separate lifetimes. We have compared this approach with established FLIM analyses such as the fitting approach[32], the phasor plot[12] and pattern matching[11](Supplementary Fig. 10). We show that the average lifetime approach is capable of unmixing two populations of fluorophores in a no FRET situation (Fig. 2). Moreover, we have shown that the HIV-1 virological synapse is enriched in CD4 as compared to the rest of the cell (Fig. 3) employing two blue fluorescent proteins (mTFP1 and mTQ2). Indeed, we have systematically compared the average lifetime approach, with fitting and the phasor plot obtaining similar results. Of note, the average lifetime approach was previously employed for a two species system where one of the two species is engaged in FRET[14]. Previously, we also quantitatively compared this method against the FLIM moments analysis (also developed in our lab) and the phasor plot in a FRET situation[33].

Of note, only the pattern matching approach was able to consider the crosstalk between channels (Supplementary Fig. 11). The unique signature of each FP in each of the three detectors was considered in PIE mode (Supplementary Table 3). In contrast, the complexity of the different absorbance and emission spectra of the FPs employed (Supplementary Fig. 12) had an impact on simple phasor and average lifetime analyses (Figs. 4 and 7). Of course, this was not the case when considering spectral FLIM employing spectral detectors;[19] but interestingly the commercial set-up utilized here was only equipped with three conventional photon counting detectors. Here, we demonstrate that the combined use of PIE and pattern matching with three detectors can unmix a very complex combination of FPs successfully (Figs. 4 and 7). When applied to such a complex sample (Figs. 4 and 7), the pattern matching approach can truly display its strengths. The multi-dimensional patterns contain all spectral (excitation laser wavelength and spectral detection channel) and lifetime information of each FP, i.e., complete decays for every spectral detection channel. Thus, the complete information present in the sample is exploited in a reliable, single-step procedure. It delivers a quantitative result showing both the localization and relative intensity contributions of all FPs in each image pixel. Moreover, the pattern matching performance can be evaluated by inspecting the fit residuals and by performing a crosstalk analysis (Fig. 4 and Supplementary Table 3).

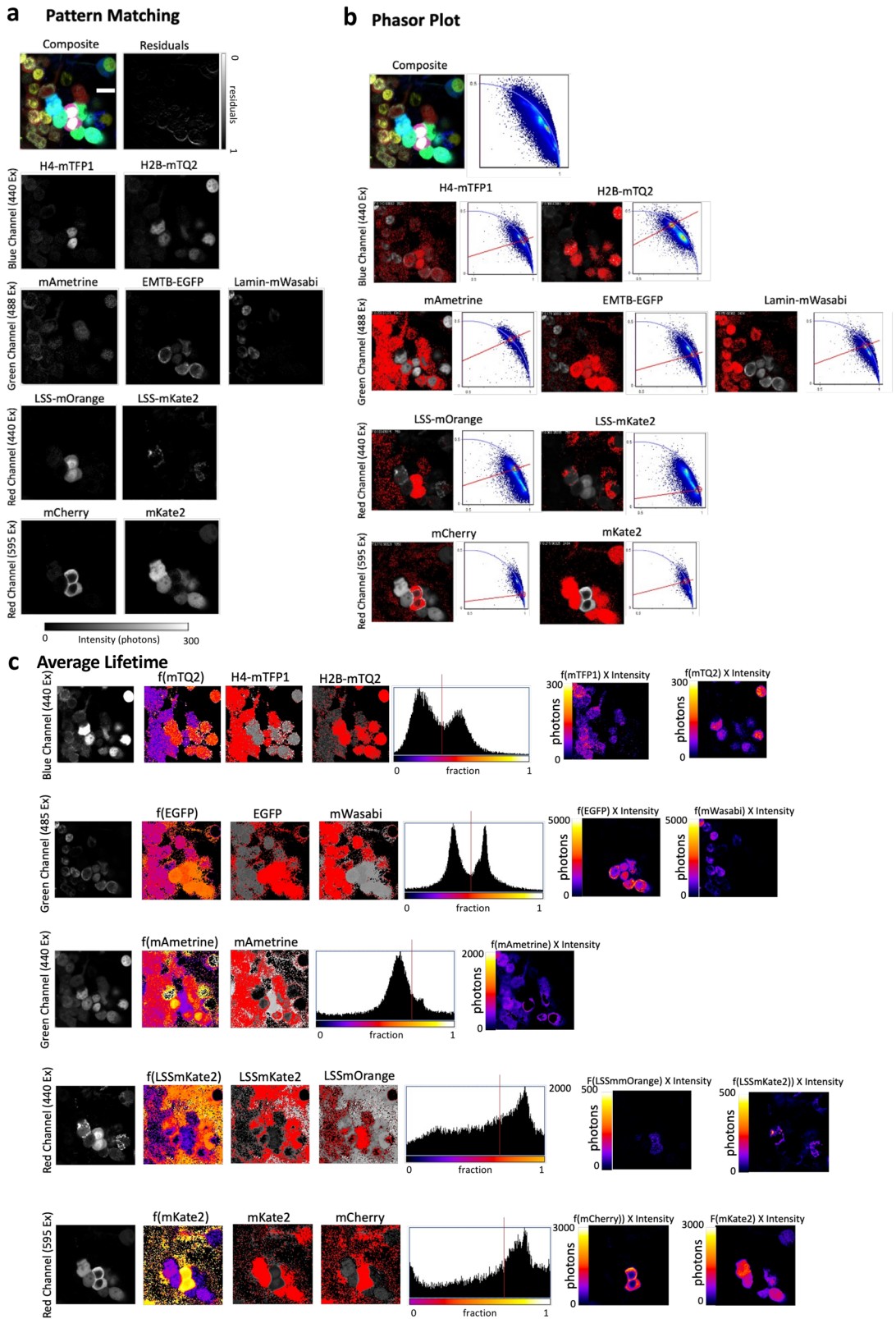

Other approaches for multicolor live cell imaging take advantage of photoswitchable FPs. The use of up to four fluorescent proteins combined with different illumination patterns has shown impressive results when expressed in bacteria and live cells combining up to four different FPs[34]. Moreover, this approach is also able to unmix 4 different FPs pixel-by-pixel. This method has also the potential of adding a second channel in red utilizing other red photoswitchable FPs (such as PA-mCherry for instance) to increase the number of colors imaged in one single shot, although this application is yet to come. It would be of interest to combine our multicolor FLIM approach with the use of different FPs (both photoswitchable and conventional) to explore how many channels one could recover.

As a biological application, we have imaged quadruple labeled HIV-1 virus and tested their functionality (Fig. 5). These particles turned

**Fig. 4 | 9 color imaging of live cells.** HEK293T cells were cultured together co-expressing up to nine different plasmids. The micrographs of live cells expressing nine different plasmids in three different spectral channels are shown (top panels). **a** Pattern Matching was applied to unmix all 9 colors. In the blue channel one has cells expressing histone H4 H4-mTFP1 and histone H2B-mTQ2 ("Ex" abbreviation means Excitation). In the green channel there are cells expressing LaminB-mWasabi, cytosolic mAmetrine and EMTB-EGFP. In the red channel one has cells expressing cytosolic LSSmOrange, Mito-LSSmKate2, Alpha-Actinin-mCherry and Cytosolic mKate2. Scale bar 10 μm. The calculated intensity micrographs depicted for each channel. The nine different fluorescent species were separated successfully. **b** The Phasor Plot is shown for the same example and following the same logic: the blue channel highlights cells expressing histone H4 H4-mTFP1 and histone H2B-mTQ2. In the green channel cells expressing LaminB-mWasabi, cytosolic mAmetrine and EMTB-EGFP are also highlighted. In the red channel one has cells

expressing cytosolic LSSmOrange, Mito-LSSmKate2, Alpha-Actinin-mCherry and Cytosolic mKate2. All Phasor plots for each channel are shown. The red circle was positioned for the S and G coordinates that correspond to the fluorescent proteins individually characterized previously. In every channel the correspondence between the Phasor Plot and the corresponding pixels in the image are shown in red. Scale bar 10 μm. **c** The average lifetime approach was also employed for this example. Here the intensity (left panels) the fraction of each fluorescent protein (third and fourth columns, highlighted in red), the corresponding proportion of two species histogram and the intensity of each species (two last panels) are shown. Every row presents a different channel commencing with the blue channel where cells expressing histone H4 H4-mTFP1 and histone H2B-mTQ2 are depicted following the same disposition as in (**a**) and (**b**). Scale bar 10 μm. Three independent experiments were performed with the same conditions.

out to be fusion competent and infectious (Figs. 5–6). Certainly, there was an intrinsic heterogeneity in the sample and a big drop in infectivity relative to the wild type virus was seen, however, the quadruple virus was still infectious. We acknowledge, however, that labeling of up to four fluorescent proteins reduced ostensibly the infectivity (Fig. 5b) and in turn this could have an impact on virus entry. The focus of the paper, however, is the development of multicolor FLIM approach and in spite of the effect of labeling on HIV-1 we could detect a sub-set of particles that was productive in terms of fusion and entry. Indeed, we could identify viral proteins such as Vpr-mTQ2 in the periphery of the nucleus of U2OS cells and Capsid-eGFP within the nucleus (Fig. 6 and Supplementary Fig. 9). We could also locate the quadruple labeled HIV-1 particles engaged in live TZM-bl cells expressing different markers for the cytoskeleton, mitochondria, and nucleus (Fig. 7); obtaining the seven independent channels with pattern matching. By quantifying the number of quadruple particles at early times right after virus exposure ($t = 0$ min) as compared to later times ($t = 90$ min) one could infer that these viruses tended to decrease in number (Gag-mCherry release) suggesting that they were indeed productive (Fig. 6). Importantly, the combination of this labeling approach together with multicolor FLIM shows that HIV-1 entry could be monitored at different infection steps, from hemifusion and fusion until capsid traffic and productive infection (Fig. 7).

In brief, our multicolor FLIM approach opens the gate to increase the complexity of imaging different biological processes, also offering the possibility to image built-in controls at the same time as other dynamic protein interactions. The use of stable cell lines and the combination with other approaches in transient transfection represents a good alternative for the limitation in the number of FPs to be expressed in single living cells. Of note, we have only focused on live cell imaging with FPs; but our approach would also be compatible with a combination of FPs and organic dyes and/or biosensors[11]; which would increase drastically the number of colors to be imaged simultaneously. Multicolor PIE FLIM combined with pattern matching analysis, therefore, opens the door for an era of live cell imaging where up to nine different color dimensions can be recovered together with time and the three spatial dimensions. We anticipate that the use of artificial intelligence in FLIM combined with non-fitting approaches[35], that require less photons to perform well as compared to fitting approaches will be the key to obtaining dynamic patterns from the huge wealth of information in such complex acquisitions.

## Methods

### Mammalian cell culture
Untransduced (UT) COS-7 or HEK293T from ATCC (CRL-1651 and CRL-11268) cells and transfected COS-7, HEK293T cells were cultured in Dulbecco's Minimal Essential Medium (DMEM) (Thermo Fisher, MA) supplemented with 1% penicillin/streptomycin, 10% FBS (Thermo Fisher, MA) at 37 °C in a 5% $CO_2$ incubator. Jurkat cells (ATCC, CRL-

2898) were cultured in RPMI (Gibco, Thermo Fisher, MA) at 37 °C in a 5% $CO_2$ incubator. All cells are routinely tested for mycoplasma (every 3–6 months). The cells were not authenticated as they were purchased from ATCC (Virginia, USA).

### Transfection
HEK293T cells, COS-7, U2OS (from ATCC, HTB-96) cells were seeded into an 8 Well Chamber ibiTreat μ-Slide (80826) at $5 × 10^3$ cells/well in 200 μl DMEM and left overnight at 37 °C. Fluorescent proteins were obtained from Addgene (https://www.addgene.org/) transfected using 0.2 μg plasmid DNA. GeneJuice (Sigma-Aldrich, Burlingdon MA, 70967) was used at 3:1 DNA. GeneJuice was mixed with Opti-MEM (Thermo-Fisher Scientific) at 50 μl/well, vortexed and left at RT for 5 min. The GeneJuice+Opti-MEM solution was then added to the 0.2 μg plasmid DNA and left for 15 min at RT. This was then added to the 8 well ibiTreat chamber slide and left overnight at 37 °C. Fusion proteins typically required a further day at 37 °C before imaging.

### Plasmid constructs
Most of the DNA plasmids used in this article were obtained via Addgene (https://www.addgene.org/; see Supplementary Table 1 and 2). hCXCR4 and hCCR5 were cloned into pmTurquoise2-C1 (#60560) vector by ligating NheI/AgeI fragments into the corresponding sites of the vector, to make hCXCR4-mTQ2 and hCCR5-mTQ2. mHoneydew and NowGFP were also cloned into the CD247 (CD3 zeta) plasmid (replacing mEos3.2)[36]

### Virus production
Gag-mCherry, Vpr-mTurquoise2, Capsid-GFP-containing, JR-FL pseudotyped viruses was carried out with transient transfection of 60–70% confluent Lenti-XTM 293 T cells seeded in T175 flasks. All DNA plasmids were transfected in Lenti-XTM 293 T cells employing GeneJuice (Novagen, Waltford, UK). Cells were transfected with 2 μg pR8ΔEnv, 1 μg pcRev, 3 μg of NL4-3 Gag-iGFPΔEnv and 3 μg of the right viral envelope and exposed to DiD. All transfection solutions were then added to cells supplemented with complete DMEM F12, then they were incubated in a 37 °C, 5% CO2 incubator. 12 h post-transfection, the medium was changed with phenol-red free, complete DMEM F12 after washing with PBS. 72 h post-transfection, the supernatant with virus particles was harvested and filtered employing a 0.45 μm syringe filter (Sartorius Stedim Biotech). Viral supernatants were concentrated 100 times employing Lenti-XTM Concentrator (Takara Bio, Clontech, Saint Germain en Laye, France) and then resuspended in phenol red-free medium, FluoroBrite DMEM (Thermo Fisher, Waltham, MA, USA), aliquoted and stored at −80 °C.

### Solutions and in-vitro assays
ATTO dyes (ATTO-Tec GmbH, Siegen, Germany) and Alexa Fluor dyes (Thermo Fisher) were dissolved at 1 mg in 50 μl of anhydrous N,N-Dimethylformamide (227056, Sigma-Aldrich) for the following

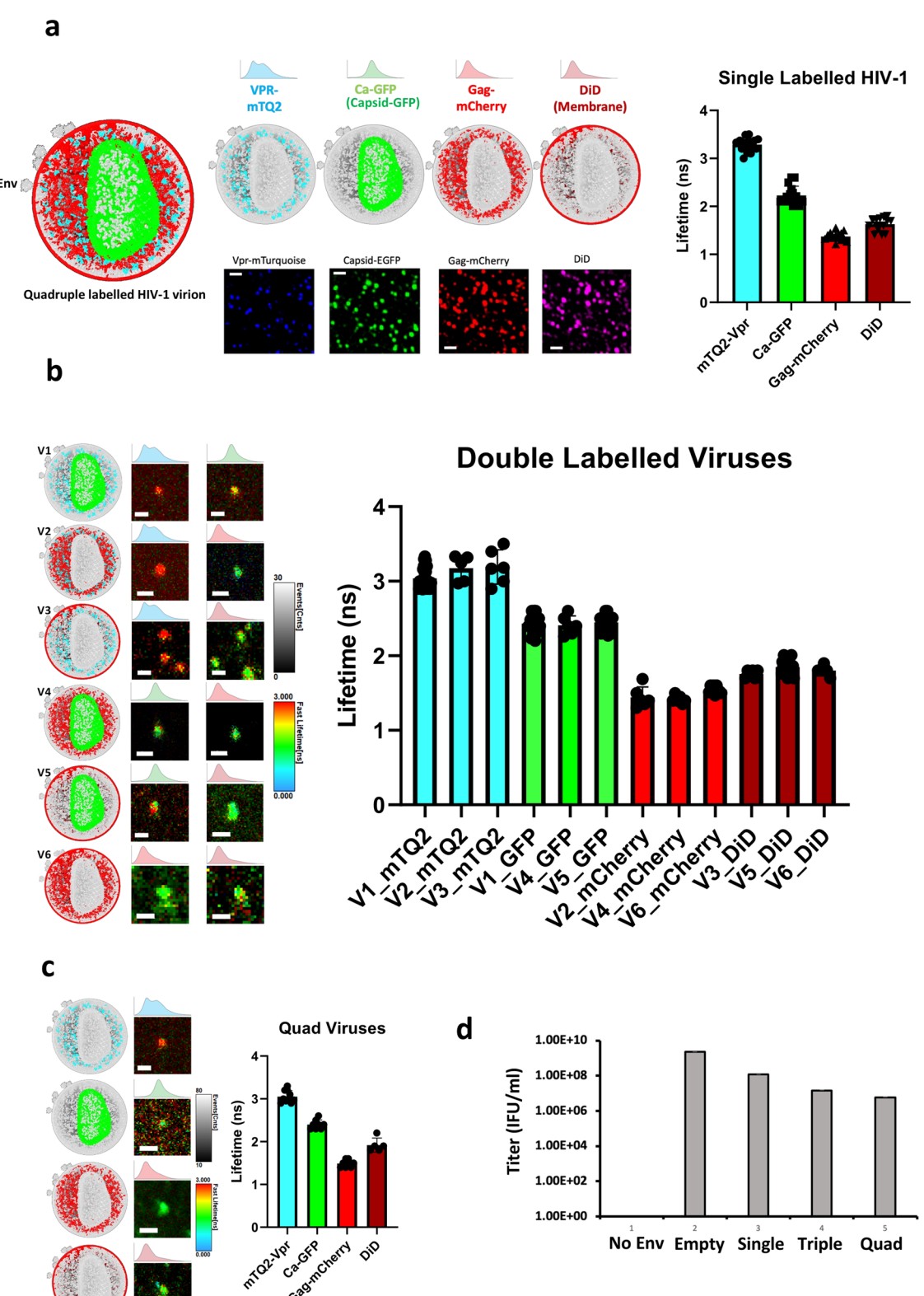

concentrations: ATTO 488 (9.37 mM), ATTO 490LS (12.22 mM), ATTO 594 (14.72 mM), and ATTO 647 N (11.52 mM), and Alexa Fluor 647 (13.33 mM). These concentrated dye stock solutions were each diluted to 10 μM in PBS. Mixtures of two dye solutions were prepared for imaging at the following combinations: 1 μM Dye 1 alone, 800 nM Dye 1 and 200 nM Dye 2, 600 nM Dye 1 and 400 nM Dye 2, 400 nM Dye 1 and 600 nM Dye 2, 200 nM Dye 1 and 800 nM Dye 2, 1 μM Dye 2 alone. Dye pairs included ATTO 488 and ATTO 490LS (Δτ ≈ 1.5 ns),

ATTO 488 and ATTO 594 (Δτ ≈ 0.2 ns), ATTO 488 and ATTO 647 N (Δτ ≈ 0.6 ns), and ATTO 488 and Alexa Fluor 647 (Δτ ≈ 3.1 ns).

A first solution containing Atto488 (Rd6G) at a concentration of 10 mM and Atto594 also at a concentration of 10 mM was prepared (in PBS). A second solution of Atto488 at a concentration of 10 mM and Atto647N at a concentration of 10 nM was prepared (in PBS). The fluorescence lifetimes of these four solutions were measured with our TD-FLIM system. We found respectively 4.02 ns for Atto488, 3.8 ns for

**Fig. 5 | Quadruple labeled HIV-1 viruses. a** HIV-1 pseudoviruses were produced with three genetically encoded fluorescent proteins labeling key viral proteins: Gag-mCherry, Vpr-mTurquoise2 and Capsid-eGFP. The fourth label was the lipophilic dye DiD. The viruses produced were tested for lifetime calculation individually. Scale bar 1 µm. The bar diagram shows the lifetime calculation for single labeled HIV-1 virions with mTQ2-Vpr (blue bar, 3.3 ± 0.1 ns, $n = 18$), Ca-GFP (green bar, 2.2 ± 0.2 ns, $n = 16$), Gag-mCherry (red bars, 1.4 ± 0.1 ns $n = 18$) and Did (dark red bars, 1.6 ± 0.1 ns, $n = 16$), where $n$ denotes number of viruses examined over 3 independent experiments. **b** Double labeled viruses were also produced and their lifetime also calculated for different combinations of labels to test whether FRET might occur between labels. Scale bar 1 µm. The different viral preps and combinations are V1, V2…V6 (for viral prep 1,2…6). The bar diagram shows mTQ2 lifetimes (cyan bars) for: V1 (mTQ2-Vpr + Ca-GFP, 3.0 ± 0.1 ns, $n = 22$), V2 (Gag-mCherry + mTQ2-Vpr, 3.2 ± 0.2 ns, $n = 5$) and V3 (mTQ2-Vpr + DiD, 3.2 ± 0.2 ns, $n = 6$); GFP lifetimes for V1 (mTQ2-Vpr + Ca-GFP, 2.4 ± 0.1 ns, $n = 22$), V4 (Ca-GFP + Gag-mCherry, 2.4 ± 0.1 ns, $n = 6$) V5 (Ca-GFP + DiD, 2.5 ± 0.1 ns $n = 17$); the red bars show the mCherry lifetime for V2 (Gag-mCherry + mTQ2-Vpr, 1.4 ± 0.1 ns $n = 5$) V4 (Ca-GFP + Gag-mCherry, 1.4 ± 0.1 ns, $n = 6$) and V6 (Gag-mCherry + DiD, 1.5 ± 0.1 ns $n = 9$); finally the dark red bars show DiD lifetime for V3 (mTQ2-Vpr + DiD, 1.8 ± 0.1 ns $n = 6$), V5 (Ca-GFP + DiD, 1.8 ± 0.1 ns, $n = 17$); and V6 (Gag-mCherry + DiD, 1.8 ± 0.1 ns, $n = 9$). $n$ denotes number of viruses examined over 3 independent experiments. **c** Quadruple labeled viruses (Quad Viruses) were analyzed to recover the different lifetimes showing that all lifetimes were consistent irrespective of the labeling strategy. Scale bar 1 µm. The bar diagram shows the average lifetime for quadruple labeled virus for Vpr-mTQ2 (cyan bars, 3.1 ± 0.1 ns, $n = 10$); Ca-GFP (green bars, 2.4 ± 0.1 ns, $n = 8$), Gag-mCherry (red bars, 1.5 ± 0.1 ns, $n = 8$) and DiD (deep red, 1.9 ± 0.2 ns $n = 5$). n denotes number of viruses examined over 3 independent experiments. **d** Infectivity assay (X-Gal assay) for different viruses labeled with one, two or four labels compared with the non-labeled wild type (positive control) and a No Env (bald particles) as negative control. All viruses were pseudotyped with VSV-G spike. The bar diagram shows infectious units per ml (IFU/ml) coming from $n = 3$ independent experiments. Data are presented as mean values +/- SD" as appropriate. Source data are provided as a Source Data file.

Atto594 and 3.4 ns for Atto647N; these values are used for a two species system with different proportions. For the lifetime of Atto488 with noise, we defocused the laser and scattered photons from the coverslip were also acquired giving more noise and an average lifetime of 4.18 ns. Next, we produced synthetic images with these two couples (Atto488 + Atto594) and (defocused Atto488 + Atto647N) with known proportions from 0 to 100% varying 10% steps. The average lifetime approach was then employed against the known percentages to evaluate the error for these ideal sollutions (Supplementary Fig. 7). Synthetic images were produced employing the weighting average formula on images employing ImageJ for each mixture (Atto488 + Atto594) and (defocused Atto488 + Atto647N). The average lifetime and standard deviation for each percentage were recovered integrating all pixels of each image.

### Infectivity assays
TZM-bl cells were used for infectivity assays. This cell line stably expresses CD4 and co-receptors CCR5 and CXCR4, required for HIV-1 infection. Moreover, they have been engineered to express the β-galactosidase gene under the control of the HIV-1 LTR promoter. Therefore, infected cells will produce the β-galactosidase enzyme capable of hydrolyze the X-gal substrate when added onto cells. This reaction will produce a characteristic blue color in infected cells that can be easily identified by transmission microscopy. One day prior to infection, $2 \times 10^4$ TZM-bl cells were seeded per well in a 96-well plate (so that cells reached ~90–100% confluency the day of the X-gal assay). Serial dilutions of the equivalent amount of physical viral particles, judged from the O.D. measurements of p24 by ELISA (Cat: SEK11695, SinoBiological, UK), were diluted in complete DMEM and added onto to cells and incubated for a further 48 h. Infected cells were then fixed using 2% paraformaldehyde (PFA). After fixation cells were washed with PBS and incubated with X-gal solution (5 mM K3[Fe(CN)6], 5 mM K4[Fe(CN)6], 2 mM MgCl2, 1 mg/mL X-gal in PBS) at 37 °C in the dark for 2 h. The X-gal solution was then replaced by PBS and cells were imaged using a Leica DMi8 microscope, Leica Microsystems (Manheim, Germany) equipped with a 10× objective. Images of each condition were obtained after merging 25 tiles. Images were analyzed using the ImageJ software (https://imagej.nih.gov/ij/). A positive signal from infected cells was highlighted to quantify the percentage of area occupied by infected cells. Similar area and threshold settings were applied to analyze each condition and the results were plotted using GraphPad Prism 9.1.0 software.

### Lifetime image acquisition
Fluorescence lifetime imaging (FLIM) was performed using the MicroTime200 (Picoquant) time resolved fluorescence microscope. The incubator was preheated to 37 °C for live cell imaging. The sample was excited using pulsed 440 nm, 485 nm, 595 nm or 635 nm diode laser (LDH series Picoquant) with a repetition rate of 20 MHz. The laser beam was coupled to an Olympus IX73 inverted microscope and focused onto the sample by a 63×, 1.2 NA water immersion objective lens (Olympus UPlanXApo). Emission for all figures used a quad-dichroic mirror specifically designed to reflect the four laser lines employed and let emission go through (Quad 440/485/594/635 nm Chroma). Detection for red photons was carried out with a SPAD (Picoquant); previously a 560 dichroic would allow red photons to access the SPAD or reflected the blue and green photons toward two PMA hybrid detectors. The blue and red photons where in turn separated with another dichroic with a 500 cut-off so that photons below this wavelenght would be directed to the blue HyD detector and photons above 500 nm (and below 560 nm) redirected to the green HyD. In front of the red, blue and green detectors, emission filters where placed (600LP, 450–480 and 500–550 nm emission from Chroma respectively). Time correlated single photon counting (TCSPC) was performed using the Multiharp 150 (Picoquant). For imaging, the size was 512 × 512 with a dwell time of 1.3 µs and pixel size of ~0.150 µm/px for 50 frames.

### Three-dimensional confocal imaging and FLIM
Automated three-dimensional confocal imaging was performed with Stellaris 5 (Leica Microsystems, Manheim Germany). Two continuous lasers tuned at 405 and 447 nm were employed to excite the DAPI and Vpr-mTurquoise2 respectively. A supercontinuum laser tuned at three wavelengths (485, 594 and 635 nm) was employed sequentially using line-interleaved excitation to avoid bleed-through to excite Ca-eGFP, Gag-mCherry and DiD respectively. A 63X (1.2NA) objective was employed, the motorized stage of the system allowed automated z-sectioning for all channels and 9 independent slices were recovered per observation field. The emission light was sent to the confocal head where the emission light would be scanned and filtered toward 5 different HyD digital detectors previously going through the pinhole that conferred a z resolution of 800 nm. The data was acquired in photon counting mode and the time-of-arrival modes turned on. This means that all channels excited with the pulsed laser would produce FLIM images that were acquired with LASX (Leica Microsystems) and analyzed with ImageJ (https://imagej.nih.gov/ij/ version 1.53c) and ICY software (https://icy.bioimageanalysis.org/) for particle detection algorithms. The, the stack of 9 images was converted into the maximal projection for intensity channels (photon counting mode) on one side and FLIM images on the other. The plugin for particle detection (https://icy.bioimageanalysis.org/plugin/spot-tracking/) was employed in both images and the statistics were recovered for the number of localized virus (with radius between 80 and 150 nm)

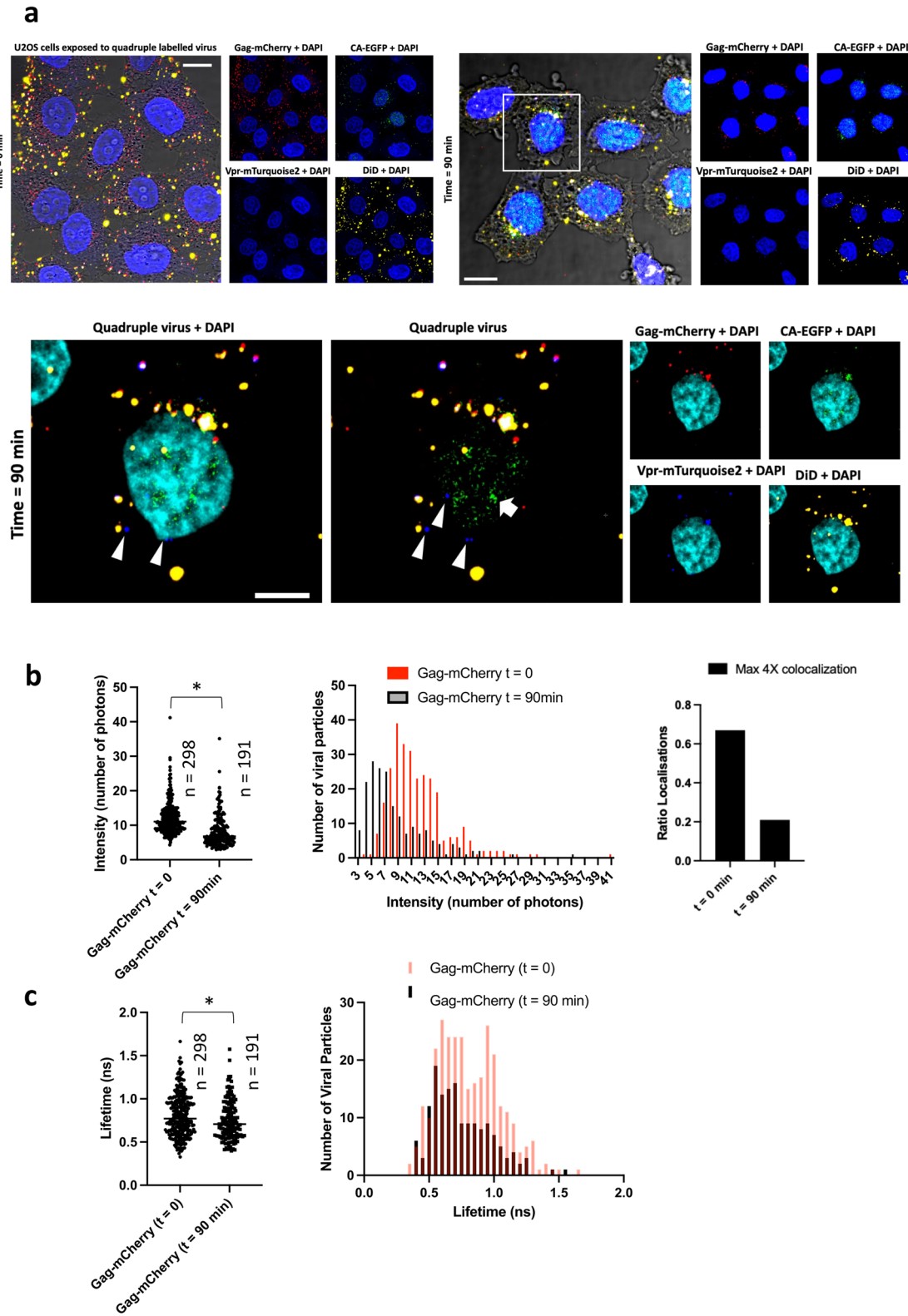

and a cut-off intensity of 30 photons. Also, using ICY the degree of co-localization was evaluated.

## Fitting of the fluorescence decays

Symphotime (Picoquant, Berlin) was employed to fit the fluorescence intensity decays employing either single or double exponential models deconvolved with the instrument response function (IRF). When using double exponential fits, the known lifetimes of the

fluorescent proteins under investigation were fixed, leaving free the rest of the parameters.

## Pattern matching analysis

The pattern-matching technique was performed as in[11]. The fluorescence decay patterns for each fluorescent protein was taken as a reference to unmix complex multidimensional fluorescence signals. Observe that in the original paper Niehorster and

**Fig. 6 | Imaging Virus entry with quadruple labeled virus. a** U2OS cells were exposed to quadruple labeled viruses (noted 4x), stained in the nucleus with DAPI and imaged at different time points ($t = 0$, left panels) and ($t = 90$ min) right panels. Merged images coming from 6 different channels are depicted: (transmission, gray, DAPI blue, Gag-mCherry red, CA-GFP green, Vpr-mTQ2, cyan and DiD, yellow) for both channels. A blow up (white square right panel, $t = 90$ min) is also depicted in the second row of micrographs. White triangles point at viral particles (Vpr-mTQ2) situated at the periphery of the nucleus. The white arrow points at capsid-eGFP within the nucleus. These particles were only seen at $t = 90$ min. Scale bar 10 μm.

**b** Quantification of HIV-1 particles with Gag-mCherry at $t = 0$ and $t = 90$ min in terms of intensity and lifetime (**c**). The corresponding histograms for both intensity and lifetime are also shown (**b** and **c** respectively). A t-test analysis was performed showing that both groups were significantly different with 0.95 confidence (*); in both cases, a t-test statistical analysis was performed with $p = 0.0129$, degrees of freedom = two tailed, and confidence intervals = −0.1039 to −0.01239. The bar diagram in (**b**) shows the quadruple labeled localization analysis (in terms of relative number of particles) for $t = 0$ and $t = 90$ min. Source data are provided as a Source Data file.

coleagues[11] employed spectral imaging with 32 spectrally separated detection channels. Here, the original data employing three different detectors and pulsed interleaved excitation was sufficient to resolve 9 fluorescent proteins. The detection probability of a given fluorophore is assumed to be the normalized probability density function $P(t,\lambda)dtd\lambda$. This function describes the number of photons per time interval $dt$ and if available the spectral information $d\lambda$. In our case each detector covers a particular spectral emission as defined above and this information can be computed as $d\lambda$ (Supplementary Figs. 4–6). When a sample containing different fluorophores is acquired both temporal and spectral information are recovered providing unique patterns:

$$I(t,\lambda)dtd\lambda = \sum n_i p_i(t,\lambda)dtd\lambda \tag{1}$$

Where $n_i$ is the amount of fluorescent protein i. Even if the exponential decay patterns and Poisson statistics of the noise prevent straightforward solution of Eq. 1, one can solve it by computing the spectral patterns summing over the temporal coordinate, note that in this case the spectral data comes from the emission patterns considering the three detectors utilized for each fluorophore (blue, orange and red).

The time-resolved photons recovered for each excitation pulse (PIE pulses) are processed in the same manner (pixels binned identically to the pattern data). The solution of Eq. 1 is found minimizing the Kullback-Leibler discrepancy as defined in Niehorster et al., 2016. A positive solution for the Kullback-Leibler discrepancy can be found utilizing the Lee and Seung algorithm[37].

The code to perform pattern matching FLIM analysis is open source and can be found at: https://github.com/PicoQuant/sFLIM

**Phasor plot analysis**
Phasor Plot analysis was performed with SimFCS software (Laboratory for Fluorescence Dynamics, University of California, Irvine). The, .bin files were uploaded into SimFCS and Phasor analysis performed with the FLIM analysis suit following Prof. Enrico Gratton guidelines (https://www.lfd.uci.edu/globals/).

The theory of the polar approach for time domain FLIM has been previously described[12]. In brief, each intensity histogram is converted into [u,v] coordinates which are respectively the cosine and sine transforms of the fluorescence intensity decay defined by:

$$u = \frac{\int I(t)\cos(wt)\,dt}{\int I(t)dt} \tag{2}$$

$$v = \frac{\int I(t)\sin(wt)\,dt}{\int I(t)dt} \tag{3}$$

**Average lifetime and unmixing two species in the same pixel: a non-fitting approach**
**Multiple exponentials.** The mean lifetime approach for FLIM is a non-fitting approach that allows calculation of the average lifetime of a particular fluorophore without a priori model and is very useful

in biology.

$$<\tau> = \frac{\int ti(t)dt}{\int i(t)dt} \tag{4}$$

Where t is the time and i(t) the intensity of the fluorescence signal at a particular delay time relative to the laser pulse. Considering a discrete sampling, the above expression can be numerically solved resulting in:

$$<\tau> = \frac{\sum \Delta t_i \times I_i}{\sum I_i} \tag{5}$$

Where $\Delta t_i$ is the time of a particular time-gated window and $I_i$ the corresponding intensity. This approach has traditionally only been applied to time-gated FLIM employing a light intensifier to generate time-gated stacks with relatively long-time gates (from 1 to 3 ns) [i.e. fast-gated CCD camera[38]]. However, this approach can also be applied to TCSPC with narrow time gates of a few picoseconds (14 ps in our case) giving rise to a more accurate measurement[39].

In the past, we have developed a method to quantify the minimal fraction of FRET for a two-species system employing the definition of the average lifetime[14]. This system can be applied to single and multi-exponential fluorophores. Here, we extend this approach to calculate the fraction of a two species model for both cases.

If one assumes a two-species model for two fluorophores with multi-exponential lifetimes then the next expression can be written to define their complex decay:

$$i(t) = f_B \sum a_i e^{-\frac{t}{\tau_{Bi}}} + (1 - f_B) \sum a_i e^{-\frac{t}{\tau_{Ai}}} \tag{6}$$

where $f_B$ is the fraction of the B species with the shortest lifetime and $a_i$ the pre-exponential factors of the different $\tau_{Bi}$, $\tau_{Ai}$ lifetime species.

By using Eq. (6) into Eq. (5) and isolating $f_B$ one can obtain:

$$f_B = \frac{[1 - <\tau>/<\tau>_A]}{\{1 - <\tau>/<\tau>_A + [((<\tau>/<\tau>_A) - (<\tau>_B<\tau>/_A))/K]\}} \tag{7}$$

where $<\tau>_A = \sum a_i \tau^2_{Di}/\sum a_i \tau_{Di}$, $<\tau>_B = \sum a_i \tau^2_{Bi}/\sum a_i \tau_{Bi}$ and $K = \sum a_i \tau_{Ai}/\sum a_i \tau_{Bi}$

Observe that all these parameters can be recovered experimentally and therefore Eq. (4) will predict the percentage of species B in a two species model in which both fluorophores behave as multi-exponentials.

**Single exponential fluorophores.** To unmix with non-fitting approaches and pixel-by-pixel the fraction of two different FPs an analytical formula can also be derived which assumes that the decays of the two FPs under consideration could modeled with a single exponential decay:

The kinetics of fluorescence decay depends on the relative proportion of the different pathways of relaxation once the fluorophores have been excited. For a single fluorophore in a homogeneous

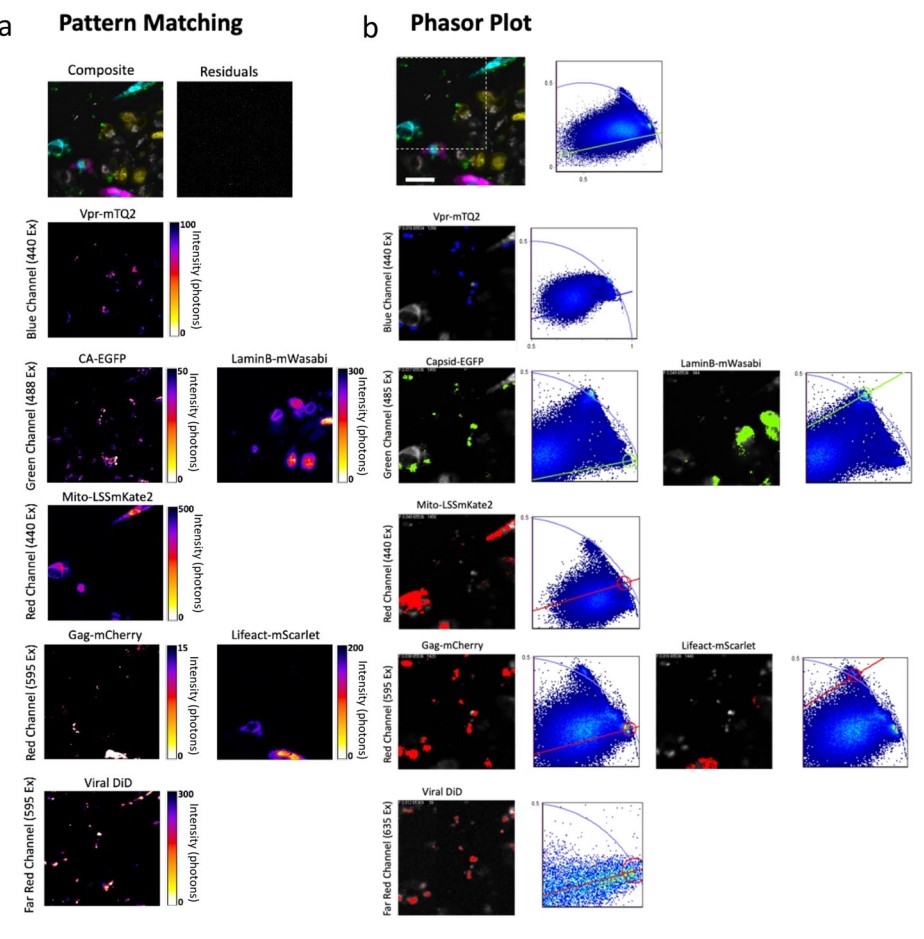

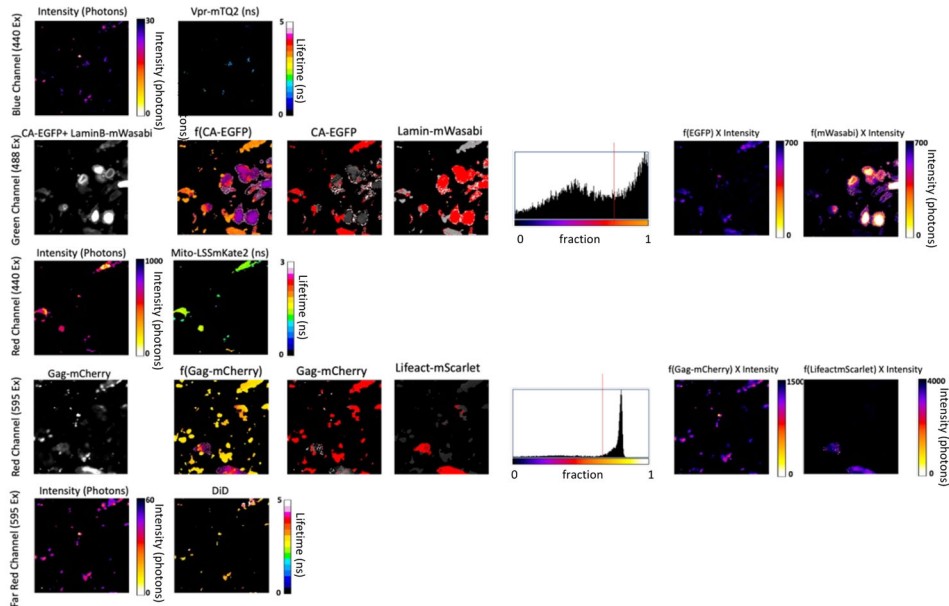

**c  Average Lifetime**

environment the decay can be described as:

$$i(t) = k_r [M^*] e^{\left(-\frac{t}{\tau}\right)} \tag{8}$$

where i(t) is the intensity as a function of time, $k_r$ the relaxation constant and [M*] the concentration of fluorophores in the excited state, t is the time and τ the average lifetime in the excited state. Now, if we consider a two-component system with two different fluorescent

proteins then

$$i(t) = k_r \left\{ [A] e^{\left(-\frac{t}{\tau_1}\right)} + [B] e^{\left(-\frac{t}{\tau_2}\right)} \right\} \tag{9}$$

A discrete double exponential describes the fluorescence decay of both FPs. If one normalizes the amplitudes of both preexponential factors to 1 we introduce the concept of the fraction of FP present in a

**Fig. 7 | multicolor FLIM imaging allows dissecting different HIV-1 infection steps during entry. a** Quadruple labeled HIV-1 particles were exposed to triple labeled TZM-bl reporter cells. Images showing the total amount of photons for all seven colors and the three detector channels merged and unmixed with pattern matching analysis are also shown. Scale bar 10 μm. **b** The same example was processed with the Phasor plot. The dotted square denotes the region subject to analysis due to limitations of SimFCS software utilized (see material and methods). The same channels are shown following the same logic as in (**a**). The two-dimensional Phasor Plots for each channel are shown and the regions for each species are highlighted in blue for the blue channel, green for the Green Channel and red for the three red channels. These regions were previously calculated for each species and highlight specific lifetimes for both viral particles and/or LaminB-mWasabi, Lifeact-mScarlet and Mito-LSSmKate2 expressed in live cells. Scale bar 10 μm. Three independent experiments were performed with the same conditions. **c** The average lifetime was employed to unmix all seven channels. The first column shows the overall intensity image for each channel. The second column the calculated fraction of fluorescent protein. The third and fourth row the highlighted fraction for each species. The fifth row the corresponding histogram for the fraction of each species and the red line is the cut off line that separates both species. The last two micrographs show the fraction of each species in terms of intensity (number of photons). Scale bar 10 μm.

particular pixel, which is similar to the one developed by us in previous work for protein-protein interactions[14]. Note, that in this case there is no Förster resonance energy transfer [FRET]:

$$i(t) = (1 - fB) \, e^{\left(-\frac{t}{\tau_1}\right)} + (fB) \, e^{\left(-\frac{t}{\tau_2}\right)}. \tag{10}$$

where $f_B$ is the fraction of the B component where we assume that the average lifetime of B is smaller than the average lifetime of A. Of course, if both lifetimes of the two FPs are equal one cannot separate their fraction as the overall fluorescence decay will be single exponential (provided that both fluorescent proteins behave as such).

If we introduce the concept of mean lifetime again, that with zero background is very similar to the photon arrival time:

$$<\tau> = \frac{\int t \cdot i(t) dt}{\int i(t) dt} \tag{4}$$

Where t is the time and i(t) the intensity as a function of arrival time.

Isolating $f_B$ from the above expression one obtains the analytical expression to be applied pixel-by-pixel to a FLIM image with two species:

$$fB = \frac{\left[1 - \left(\frac{\tau}{\tau_A}\right)\right]}{\left[1 - \left(\frac{\tau}{\tau_A}\right) - \left(\frac{\tau_B}{\tau_A}\right)^2 + \left(\frac{\tau}{\tau_A}\right) \cdot \left(\frac{\tau_B}{\tau_A}\right)\right]} \tag{11}$$

The above expression offers the possibility to obtain the fraction of the species pixel by pixel (Supplementary Note 1) with the shortest lifetime. To obtain the complementary fraction is trivial:

$$fA = 1 - fB \tag{12}$$

**Calculation of photon contribution of each fluorophore species**
Once the fraction of the fluorophore is obtained with fitting, average lifetime or the phasor approach one can recover the contribution of each species in terms of relative number of photons multiplying this fraction by the total intensity pixel-by-pixel. For example:

$$I_A = f_A \, x \, I_t \tag{13}$$

If the two fluorophores under consideration have different brightness, $I_A$ needs to be multiplied by the ratio of both:

$$I_A = f_A \, x \, I_t \, x \, \frac{B_A}{B_B} \tag{14}$$

where $B_A$ and $B_B$ are the brightness for species A and B respectively. This way, one can recover the pixel-by-pixel contribution for each species correcting for brightness.

## Statistics and reproducibility
For FRET-FLIM analyses, calculation was performed with SymPhoTime software (Picoquant, Berlin) and ImageJ (https://imagej.nih.gov/ij/). Mean and standard deviation for each individual experiment were calculated using GraphPad Prism v9.1.0 software. The sample size and the number of experiment replicate performed are specified in each figure legend.

## Reporting summary
Further information on research design is available in the Nature Portfolio Reporting Summary linked to this article.

## Data availability
The data supporting the findings of this study is available at https://osf.io/gtsud/. The FLIM data generated in this study can be accessed at https://osf.io/gtsud/. The FLIM data are available under restricted access due to the large size of these files (hundreds of MB), access can be obtained by contacting the corresponding author. Requests will be filled within 4–5 weeks. The raw data are protected and are not available due to data privacy laws. The processed data are available at https://osf.io/gtsud/. The data generated in this study are provided in the Supplementary Information/Source Data file. The data used in this study are available in the osf database under accession code https://osf.io/gtsud/. Source data are provided with this paper.

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

## Acknowledgements

We thank Dr. Vinay K. Pathak for the kind gift of the HIV-1 Capsid-eGFP plasmid. We thank all members of the Padilla-Parra lab for valuable discussions and criticism of the paper. We also thank Dr. Luis Alzarez from Leica Microsystems for technical support. This work has been supported by the European Research Council (ERC-2019-CoG-863869 FUSION to S.P.-P.), The Chan Zuckerberg Initiative "Multi-color single molecule tracking with lifetime imaging" (2023-321188 to S.P.-P.) and the Wellcome Trust Core Award (203141).

## Author contributions

Conceptualization, S.P.-P.; methodology, T.S, I.C.-A., D.J.W and S.P.-P.; validation, T.S, M.L.-K., B.K. formal analysis, M.L.-K., B.K., T.S., D.J.W and S.P.-P.; resources, S.P.-P.; data curation, D.J.W. and S.P.-P.; writing-original draft preparation, S.P.-P.; writing—review and editing, T.S., I.C.-A., M.I., D.J.W. I.C.-A., and S.P.-P.; visualization, T.S., D.J.W. and S.P.-P.; supervision, S.P.-P.; project administration, S.P.-P.; funding acquisition, S.P.-P. All authors have read and agreed to the published version of the paper.

## Competing interests

The authors declare no competing interests.
