## [Peer Review File · Nature Communications]

REVIEWER COMMENTS

Reviewer #1 (Remarks to the Author):

In their manuscript, Starlin et al propose an approach for “easy-to-use” multicolor fluorescence lifetime imaging (FLIM) to distinguish up to nine fluorescent proteins simultaneously, relying on both their emission/excitation spectra and fluorescence lifetimes. Subsequently, the authors employ the multiplexed confocal FLIM approach to visualize quadruple labelled HIV-1 viruses during infection of cells, in which four different organelle compartments are also labelled by FPs. As FLIM is a technology still not having reached its full potential, the present work nicely showcases the method, in a biologically relevant context. This is further supported by the fact, that the employed methodology for confocal multicolor FLIM is well known and commercially available, i.e. readily accessible to the research community. However, I have serious reservations with respect to the proposed non-fitting evaluation approach used to identify appropriate combinations of FPs for multiplexed FLIM and also, later on, used to resolve the FPs in the multicolor FLIM images.

The benefits of the proposed approach over already existing approaches for multiplexed FLIM (based on the phasor-approach to FLIM) remain elusive, as a direct comparison is missing. The effects of several assumptions of the proposed non-fitting algorithm on the result accuracy are not discussed at all, although they should have a direct impact on the selection of suitable FP combinations and on resolving the FPs later on in cells.

Without the clarification of these aspects, it is difficult to appreciate the added value of the work from the FLIM-methodological point of view.

Major concerns:

Whereas the evaluation approach described by Scipioni et al was applied to spectra FLIM (32 detection channels), it is applicable and has been applied also for different numbers of emission/excitation channels, as proposed for multicolor FLIM-based spatial transcriptomics (Vu et al, Nat Commun 2022). Thus, a direct performance comparison between the here-proposed non-fitting approach and the phasor-approach is necessary, to provide information on the added value of the here described algorithm. This kind of comparison would be particularly important, as the phasor approach, in contrast to the here proposed approach, is a model-free approach of FLIM data representation. Thus, representing the FLIM data in the phasor plot is not affected by deviations from the mono- or bi-exponential behavior of the fluorescence decays, e.g. caused by detector noise, whereas the here proposed non-fitting algorithm relies on a pre-defined bi-exponential model. Such situations occur especially at low photon budgets, typical for biological samples, as correctly pointed out also by the authors.

It would be important for the non-fitting fluorescence decay evaluation model to name the assumptions it makes and the impact they have on the selection of FP combinations and/or resolving certain combinations of FPs within cells. One aspect it is addressed in the manuscript: the influence of the photon budget on the accuracy of the result have been tested in comparison to mono-exponential fitting. Several other assumptions with impact on the FLIM data evaluation are not discussed. These need to be thoroughly tested, to provide a good validation of the model and to verify the choice of FP combinations for multicolor FLIM.

The model does not take into account the limited time window of the measurement, but assumes it to be infinite (deriving Eq. 5 from Eq. 3 and 4). Whereas for very short fluorescence lifetimes this assumption holds true, the longer the fluorescence lifetime turns, the less accurate the model will be. This leads to a loss in result precision for longer lifetimes. I suggest, similar to the table in Fig. 1a, to show a quantification of the precision to resolve between the FPs, with respect to the ratio of their fluorescence lifetime to the time window of the respective measurement. Related to this aspect: was the measurement time window kept constant for all experiments or was it adapted, for the different wavelengths employed for PIE or, eventually, depending on the fluorescence lifetime of the FPs used in the experiment?

Both background and electronic noise of the detector impact on the accuracy of the result in FLIM evaluation. The model considers a background-free situation (Eq 1,2,3). Therefore, I suppose the background was subtracted from the measured signal, before processing the data using the non-fitting approach. How was the background value determined: in each image, at each timepoint after the laser pulse? As the background distribution should be a Poisson distribution (eventually a super-Poisson distribution), it is asymmetric. Hence, subtracting its mean value from the signal

would affect in a different way fluorescence decays with shorter than with longer lifetimes. Therefore, it would be important to have a description of the steps performed prior the non-fitting model is applied.

The instrument response function (IRF) should be shown, as the model considers it to be a delta-function. I expect the IRF to be narrow, however, its effect on shorter as compared to longer lifetimes should be demonstrated.

Specific Comments:

Eq. 5 in the description of the algorithm for should read $\langle \tau \rangle = ((1-fB) \cdot [\tau_A]^2 + fB \cdot [\tau_B]^2) / ((1-fB) \cdot \tau_A + fB \cdot \tau_B)$, and not $\langle \tau \rangle = ((1-fB) \cdot [\tau_A]^2 + fB \cdot [\tau_B]^2) / ((1-fB) \cdot [\tau_A] + fB \cdot \tau_B)$. The final equation is however correctly derived, so that the results shouldn't be compromised.

Reviewer #2 (Remarks to the Author):

Starling et al. describe a method to unmix simultaneous imaging of nine fluorescent proteins. They then use this to investigate HIV infection, first looking at the infectious synapse and then early replication.

The microscopy technique is out of my area of expertise and so I cannot comment on how well it works or what advantage it has over existing methods. However, I can say that it is quite difficult for a non-expert to follow the figures and understand what is being measured and calculated. A summarised simple explanation in the text would help.

Regarding the use of this technique to study HIV biology, in general it is hard to assess the progression of infection as many of the controls are missing (for example, comparisons with unlabelled and individually labelled virions). I appreciate that the authors are trying to recapitulate published results, but I think they should control for their visualisation. Additionally, only one third of the particles were quadruple labelled, which might confound any conclusions someone using this technique could draw. This is not necessarily a fault with the imaging technique, but more a general issue with labelling viral particles. However, considering the controversy in the field over using the CA-GFP fusion (particularly over how much CA is labelled and what this fraction represents) and with whether the Vpr fusion localises with cores, there may be too much uncertainty with these labelling techniques (regardless of what label is used or how it is analysed) to make this really useful for understanding HIV biology at this time. Not to mention the particle to infectivity ratio that implies most of the particles visualised are defective. Perhaps using an integrase label or following viral nucleic acid and watching particles colocalise with cellular markers would be more productive?

Specific points:

Figure 3: The authors need to show singularly labelled cells expressing only CD4 or CXCR4 so that the reader can see what each signal looks like and then compare that to the digitally separated image in the co-expressed particles. It would help if the outline of the virus producing cell was indicated. They should also compare their synapse to a negative control ie with no synapse (using no env expression). Does the CD4 move? This really needs a comparison with a negative control or a time course in order to make sense.

Figure 3: The figure legend wording is not clear and doesn't match the figure. Panels should be labelled A, B, C etc for clarity. The bar diagrams are confusing and the bottom left and bottom right graphs are not even mentioned. What does "proportion" mean? How is this calculated?

Figure 5a: The diagram does not show clearly where the DdD labels. The viral titre is meaningless unless there is a comparison with unlabelled virus. The authors should also show the titre of viruses singularly labelled with individual fusion proteins to see how each one affects viral infectivity. Are the images in this figure showing viruses on coverslips (i.e. not in infected cells)? This should be made clear

Figure 5b: The line profile doesn't appear to show quadruple labelled virus? The percentage co-localisation graph should show all the possible combinations of labels, ie how many particles were singularly labelled and each pairwise combination (even if some combinations had zero particles) to see the labelling efficiency. I would expect all particles to be Gag-mCherry and DiD positive if these are particles on coverslips? Therefore, I'm not sure that "double labelled" is the correct term for particles also containing CA-GFP?

Figure 5c: Do the images show particles produced with only two labelled constructs or just show the detection of two labels? This should be clarified.

Figure 6: I don't think we really know what a Vpr or CA signal on their own actually represent? Likewise, it is stated in the legend that particles with three colours (not DiD) represented fusion, but then shouldn't most of the mCherry also disappear? Can you measure the decrease in amount of mCherry staining? If it isn't lost, then it is even more unclear what the single labelled puncta are? As it stands, this analysis is not very useful as we cannot be sure what the staining means, especially as only a third of the particles start off with all four labels?

Figure 6c: When it says "Single viruses for each label" does this mean particles made with just one label or detected on one channel? Please make this clearer throughout the paper.

Minor points

Line 145: It states "the different percentages of colocalization were quantified in Figure 5a" - this is incorrect. I think it should be Figure 5b?

Line 159: It would be helpful to state here what the lifetime of mTurquoise2 is in cells on its own.

Line 205: I don't think virus producing cells normally have abnormal nuclei?

Reviewer #3 (Remarks to the Author):

The manuscript of Starling et al demonstrates the power of combining lifetime and spectral information to multiplex the imaging of fluorescent proteins. Using their simplified approach, they have simultaneously measured nine different signals. They then apply their method to visualize the uptake and egress of HIV-1 virus like particles. Overall, the work has potential but some explanations and clarifications as needed before the manuscript can be considered for publication.

Major points:

1) The main deficiency in the current manuscript is the oversimplification of their approach and insufficient characterization of the method.

i) First of all, what is the relaxation constant, k_r ? Is this $1/\tau$? Is this a proportionality constant to convert decay events into collected photon intensity? Equation 1 is referring to "a single fluorophore", although I assume the authors mean a single type of fluorophore. It gets more confusing when the same k_r is used in Equation 2 for two types of fluorophores. Please clarify.

ii) If you are describing the intensity, the brightness of fluorophore plays a role (determined by the probability of excitation, of detection and the quantum yield of the fluorophore). While the detection efficiency may be similar for dyes with similar spectra, the other terms will not be the same in general and the relative intensities will depend on their relative lifetimes. I understand the authors try to utilize fluorophores of similar brightness, but they should expand their calculations to correct for differences in brightness. This is straightforward and does not add to the complexity of the analysis when the information is already known. They should also discuss this in more detail.

iii) Also, would it be possible to correct for FRET?

iv) As the authors are using an analytical expression for calculating the ratio of lifetimes, they will always get a result. However, they authors should investigate how closely this represents the real situation and discuss the limits. (This will be discussed in more detail below). In FLIM, a factor 2 in lifetime is usually need to reliably separate two species. Of course, if you assume two lifetimes,

the algorithm will give you a result, but how significant is it (e.g. in distinguishing eGFP ($\tau = 2.4$) and mWasabi ($\tau = 3.1$ ns)). How does the reliability of their method depend up on the difference in fluorescence lifetime?

v) The authors should also check the reliability and robustness of their approach to that of fitting each pixel to a biexponential decay (or to the phasor approach).

2 Section: "Selecting FPs...". The authors discuss that having an exponential decay is important for their approach and they investigated 30 available FPs. It would be useful if the authors were present the exponential decays of the 30 FPs they investigated in the SI with mono-exponential fits and residuals so that the reader can get an idea of how mono-exponential the different FPs are.

3 Section: "Pixel by pixel unmixing...":

i) Here, there is no known ground truth. In addition, for the tandem, I would expect FRET. Have the authors checked for FRET. What are the brightnesses of mTFP1 and mTurquoise2 under these conditions? What excitation powers were used (this information should be given somewhere!). In addition, one does not know the maturation of the different proteins and transfection efficiencies when using different plasmids.

ii) To verify their methodology, the authors should artificially mix "pure" images to make a combined simulated image where the expected results are known and compare the results of their analysis. In addition, they should vary the contributions of the two species to explore the limits of their approach.

4. Section: "Defining the proportions of HIV-1 receptors..."

Here are the first potentially interesting results. However, a lot of the useful information that could be extracted from such measurements is missing. i) What is the distribution of CD4 to CXCR4 expression in cells not exposed to viruses. ii) Does this correlate with the total concentration of CD4 or CXCR4? iii) The spatial distribution of viruses in Figure 3 is very inhomogeneous. Can the authors explain this? iv) In the presence of viruses, are their results significantly different than the 80%/20% you would expect? v) Does this ratio also vary with total concentration of the receptors available?

5. Section: "Simultaneous Imaging of nine FPs..."

How do the authors know that what they measure is correct? Clearly the algorithm will give a result. Is there some other feature that one can be used to verify that the determined species is really the species of interest? (e.g. H2B will make dots in the nucleus, a plasma membrane marker will be clearly in the periphery, etc). In Figure 5, panel c, it appears as if there is a clear segregation of the different plasmids with only one plasmid present per cell. I would expect most cells to have incorporated all plasmids they have been exposed to, and most likely with similar ratios. Please explain why this is not the case.

6. Section: "Single particle detection of quadruple labelled HIV-1..." Lines 197-199: "The second subpopulation of viruses engaged in the cells was further analyzed to unmix the viral and cellular compartments and detect different steps of the HIV-1 infection cycle." How is this done and where are these results?

7. Figure 7 does not contribute new information to the work and can be removed. This is a summary of things already known.

8. There are many relevant studies that have combined FLIM and spectral imaging, or used fluorescence lifetime to investigate viruses that should be cited within this paper. The citations are very egocentric, focusing on work from the Padilla-Parra lab and not on the general literature.

Minor points:

1. Lines 43 and 260: The authors twice mention artificial intelligence, but other than throwing out the name, there is not substantiation of how they expect AI to contribute. This should either be more appropriately discussed with proper references in the discussion/outlook or removed from the paper.

2. Lines 49-51: As the authors know, the necessity of a single exponential decay is not necessary when analyzing with the phasor approach. The authors should mention this in the introduction.

3. Line 208: "Nucleoplasmic"

4. Line 394, Equation 6: τ should be $\langle \tau \rangle$ in the formula.

5. Figure 1: Panel a, right side. I do not understand these tables and what is being shown. e.g. the upper left box for blue excitation has a lifetime of 1.9 ns and a label of mTFP1 on top and mTQ2 on the left. What does this lifetime refer to?

6. Figure 2: i) Using a) b) and c) for panels and to simultaneously to discuss parts of a single panel is confusing. Please use a different notation within a panel (e.g. i, ii, iii...). ii) What is being shown in the middle sub-panels of panel a. Information/labels and description are missing here.
7. Figure 5, Panel a, right side: The X-gel assay should be better described. I do not understand what is being shown in this figure (I can only guess).
8. Figure 6, Panel a, right side: The arrows are confusing. Better would be data points. Please also plot the number of viruses for viruses on the coverslip (so one can see that they remain constant).
9. Figure 6, Panel b, fourth row: There are cells visible in the FLIM image that are not described as subpopulations. I assume this is cross talk of mScarlet into the red channel. However, the authors need to explain how they deal with this and not just ignore it. As often in the paper, important details are missing.
10. Line 604: Supplementary Figure 4: "and their errors increase from 200 photons per pixel". I have never heard before of errors in a fit increasing when the number of photons increases. Please reword.

RESPONSE TO REVIEWERS' COMMENTS

Reviewer #1 (Remarks to the Author):

In their manuscript, Starling et al propose an approach for “easy-to-use” multicolor fluorescence lifetime imaging (FLIM) to distinguish up to nine fluorescent proteins simultaneously, relying on both their emission/excitation spectra and fluorescence lifetimes. Subsequently, the authors employ the multiplexed confocal FLIM approach to visualize quadruple labelled HIV-1 viruses during infection of cells, in which four different organelle compartments are also labelled by FPs. As FLIM is a technology still not having reached its full potential, the present work nicely showcases the method, in a biologically relevant context. This is further supported by the fact, that the employed methodology for confocal multicolor FLIM is well known and commercially available, i.e. readily accessible to the research community.

Thanks to the reviewer's positive comments on our effort to visualize nine fluorescent proteins simultaneously and the application of this technology on virology. We agree that FLIM has not reached its full potential and that our work would be accessible to a broad research community as it employs commercially available instrumentation.

We have now validated our average lifetime approach and extend its application to fluorophores with complex decays better defined as multiple exponentials. We have also compared this method together with the fitting approach and the phasor plot.

We also acknowledge the limitations of this approach (together with the phasor plot and the fitting analysis) in the context of multiple fluorophores such as nine color FLIM imaging. This is why we have introduced pattern matching FLIM analysis. Importantly, we have applied this method without the need of having an array of spectral detectors as previously done by [Niehorster et al., 2017].

The conclusions of the paper have slightly changed as compared to the previous version of the manuscript: first, we present the non-fitting approach, which is valid in simple two species systems, showing a biological application (the virological synapse, New Figure 2-3). Second, we introduce pattern matching for complex systems with several fluorophores. We were able to show that it is possible to image in one single shot up to nine fluorescent proteins considering the degree of crosstalk for each channel for the fluorescent proteins employed in live cells. Moreover, we compare this approach with both the phasor plot and the average lifetime and discuss its implementation in live cells and also in the context of quadruple labelled HIV-1 entry.

We believe we have answered all queries and that the manuscript has improved ostensibly as compared to the first submission.

However, I have serious reservations with respect to the proposed non-fitting evaluation approach used to identify appropriate combinations of FPs for multiplexed FLIM and also, later on, used to resolve the FPs in the multicolor FLIM images.

We understand that this non-fitting approach deserves careful explanation and hope that the changes, new data and detailed explanations throughout the new version of the manuscript will help to convince the reviewer that our approach is indeed accurate.

We would like to stress that our non-fitting approach was first developed and applied to a two species system with both single and multiple exponential fluorophores more than a decade ago (Padilla-Parra et al., 2008 Biophys J). Back then, we were applying the average lifetime non-fitting approach for different FRET scenarios, always with a two species model. Ever since, we have employed this approach in many biological examples [for instance: Yamada et al 2009 JBC; Zhao et al, 2014 Nat Comm, Kong et al., 2016 Neuron]. In fact, our method and the phasor plot (Digman et al., 2008) came up simultaneously.

It is important to understand, that we are now employing the same approach to unmix two different species that do not undergo FRET necessarily; with either single or multiple exponential behaviour (see new theory Material and Methods). We have now included the mathematical description for the two species system with fluorophores with multiple exponential behaviour and have applied this approach also in comparison with both global fit analysis and phasor plot approaches. We acknowledge that the non-fitting average lifetime is mostly useful in a rather simple system where two species with different lifetimes are considered. We have compared this approach with the phasor plot and fitting approaches obtaining similar results.

For multicolor FLIM we show in the new version of the manuscript that pattern matching [Niehorster et al., 2017 Nat Methods; Rohilla et al., 2019, Scientific Reports] is more accurate as compared to phasor plot and average lifetime analyses (New Figure 4 and 7).

The benefits of the proposed approach over already existing approaches for multiplexed FLIM (based on the phasor-approach to FLIM) remain elusive, as a direct comparison is missing.

In the previous version of the manuscript we had introduced the non-fitting approach to unmix a population of two fluorophore species with single exponential behaviour; and had compared this with fitting approaches (mTFP1 and mTurquoise2, Figure 2) and also for other two species with different excitation and detection spectra (Supplementary Figure 1). We have now included fitting approaches (double exponential fit) and the Phasor Plot in all cases to validate the Average Lifetime approach. (New Figure 2b-c, New Figure 3,4 and 7 and New Supplementary Figure 1 and 2). We also compare our approach with the Phasor Plot and Pattern Matching in (New Figure 4 and New Figure 7).

We would like to stress, however, that the analytical solutions derived here based on the definition of the average lifetime employing a non-fitting approach in the context of FRET had already been validated in our seminal paper (Padilla-Parra et al., 2008) for a two species model with one species undergoing FRET (for both single and multiple exponential donor

decays). Further work by our lab also compared side by side this approach with the phasor plot and a novel moment FLIM analysis also developed by us (Leray et al., 2013). These papers were referenced in the first version of the manuscript and we have now discussed this thoroughly in the current version [L106– L124].

In the new version of the manuscript we have arrived to the conclusion that the best approach to reliably unmix 9 FPs is pattern matching. The reliability of the pattern matching algorithm was also previously evaluated in [ref. Niehörster et al. 207]. In that article the authors compare pattern matching to other algorithms for FLIM, in particular phasor analysis [Figures 8 to 14 and Supplementary Note 1. Supplementary Note 2]. We also do this in our New Figures 4 and 7.

The effects of several assumptions of the proposed non-fitting algorithm on the result accuracy are not discussed at all, although they should have a direct impact on the selection of suitable FP combinations and on resolving the FPs later on in cells. Without the clarification of these aspects, it is difficult to appreciate the added value of the work from the FLIM-methodological point of view.

We thank the reviewer for this insightful comment. It is true that a number of assumptions were taken in order to apply the non-fitting algorithm. Thanks to the comparison against the phasor plot, pattern matching and fitting procedures deconvolved with the experimental IRF with single and double exponential models we are now discussing the strengths and weaknesses considering new results for all FLIM data presented.

We demonstrate that the approach based on the average lifetime considering a two species model is applicable and comparable in all cases. We also evaluate the error of our approach (as suggested by reviewer 3, see below) in a system with different mixtures of solutions (New supplementary figure 3). We show that the approach is only limited by the intrinsic error of the measurement and the signal to noise. It shows the same trend as in our old publications [Padilla-Parra et al., 2008 and Leray et al., 2013]. Observe, that in this case instead of a FRET system with two species we have a no FRET system with two species. This makes the system much easier to resolve as both lifetimes can be recovered experimentally (with independent experiments). In the case of FRET we developed the concept on the minimal fraction of interacting donor (mFD) which was recovered also with non-fitting approaches where TauFRET does not need to be known a priori [Padilla-Parra et al., 2008. Biophys J]. We also discussed these non-fitting approaches side by side in Padilla-Parra et al., 2011 Biophys Rev (not cited in the current manuscript).

Major concerns:

Whereas the evaluation approach described by Scipioni et al was applied to spectra FLIM (32 detection channels), it is applicable and has been applied also for different numbers of emission/excitation channels, as proposed for multicolor FLIM-based spatial transcriptomics (Vu et al, Nat Commun 2022). Thus, a direct performance comparison between the here-

proposed non-fitting approach and the phasor-approach is necessary, to provide information on the added value of the here described algorithm.

Thanks for this suggestion. We have now employed the Phasor Plot in all examples to compare our non-fitting approach with the Phasor FLIM. It is worth noting that we already published a paper where these two methods were exhaustively compared for a two species system (one engaged in FRET and one unperturbed; Leray et al., 2013.). We understand, however, that further validation was needed in this new application and have followed the reviewer's suggestions for all FLIM data (New Figures 2,3,4,7 and Supplementary Figures 1-2). Moreover, we have also included the pattern matching approach [Niehorster et al., 2016. Nat Methods] that takes into consideration the unique spectral signature of each fluorophore and its impact in each one of the three detectors [New Figure 4 and Supplementary figures 5-8]. We discuss the patterns obtained in all cases and their similarities and accuracy in the discussion section.

This kind of comparison would be particularly important, as the phasor approach, in contrast to the here proposed approach, is a model-free approach of FLIM data representation. Thus, representing the FLIM data in the phasor plot is not affected by deviations from the mono- or bi-exponential behavior of the fluorescence decays, e.g. caused by detector noise, whereas the here proposed non-fitting algorithm relies on a pre-defined bi-exponential model. Such situations occur especially at low photon budgets, typical for biological samples, as correctly pointed out also by the authors.

One could roughly say that the average lifetime calculation is the equivalent of the Phasor approach considering light as photons (one dimension) instead of a wave (phase and modulation, two dimensions). In both cases they are model-free approaches that necessitate the assumption of a two-model system if one needs to separate two different fluorophores with different lifetimes. This assumption in the case of the average lifetime in one dimension gives rise to a two-model system where the two species might behave as multiple exponential fluorophores or not. We have derived these equations and show that our model also considers species that might behave as a multiple exponentials as well. The single exponential case is just a particular case of this one.

We agree with the reviewer and indeed our non-fitting approach was based on a two species system where each fluorophore presents a single exponential behaviour and it is assumed that the contribution of each species can be derived provided they present a high number of photons and a good signal to noise. In the past, we also applied this system for multiple exponential fluorophores, also considering a two species system where one species is engaged in FRET (Padilla-Parra et al., 2008). We have also introduced the mathematical formulation for this scenario adapted to a NO-FRET example and we further compare this approach with the Phasor Plot. In both cases these FLIM analyses perform well in a simple case scenario (New Figure 2 and 3). The new data produced also compares the performance of both phasor plot and average lifetime in a more complex situation (multicolor FLIM) showing that only pattern matching was able to unmix correctly all nine colors (New Figure 4).

Of note, the pattern matching approach does not make any assumptions beyond the very basic one that the fluorophores behave the same in the single labeled reference sample and the multi-color sample. It records the shape of the decay without evaluating if it shows a single- or multi-exponential behavior. Therefore it can also be called a model-free approach. Moreover, the IRF is also included in the shape of the decay. All reference patterns are used in the linear unmixing step. In case that a label is not present, the corresponding amplitude would be zero. Spectral bleed-through is not a concern in the sense that the intensity distribution in all analysis channels is contained in the pattern and is evaluated during the analysis. Similarly, the background is included in the pattern.

The conclusions of the paper are now that both phasor plot and average lifetime are able to unmix simple systems such as two species system. However, more complex scenarios such as 7 and 9 multicolor FLIM approaches necessitate pattern matching. Observe, that we have employed the same set up which is not equipped with spectral FLIM, however, we could still recover the proportion of each species with this analysis.

It would be important for the non-fitting fluorescence decay evaluation model to name the assumptions it makes and the impact they have on the selection of FP combinations and/or resolving certain combinations of FPs within cells. One aspect it is addressed in the manuscript: the influence of the photon budget on the accuracy of the result have been tested in comparison to mono-exponential fitting. Several other assumptions with impact on the FLIM data evaluation are not discussed. These need to be thoroughly tested, to provide a good validation of the model and to verify the choice of FP combinations for multicolor FLIM.

We thank the reviewer for this suggestion. We have performed a new set of experiments to evaluate the impact of the non-fitting approach on single fluorescent proteins (New Supplementary Figure 1), comparing two species mixtures of fluorescent proteins (New Figure 2 and New Supplementary Figure 1 and 2) and also when applying our PIE approach to 9 different fluorescent proteins (New Figure 4) and when applying this employing multi-labeled viruses (New Figure 7). All these examples show that the average lifetime approach coincides well with the Phasor Plot and also with fitting approaches (provided with have enough photons in this last case). These comparisons are also discussed in New Figure 4 and 7 and in the text now (L170-L182).

Moreover, we have now concluded that only the pattern matching approach was able to take into account spectral crosstalk and give a reliable solution in complex multicolor situations. To evaluate the performance of the pattern matching analysis, the residual crosstalk between the unmixed images has been calculated as described in

: Winter, Franziska R., et al. "Multicolour nanoscopy of fixed and living cells with a single STED beam and hyperspectral detection." *Scientific reports* 7.1 (2017): 1-11. DOI: 10.1038/srep46492

The relative brightness values of the different labels influence the crosstalk result [Supplementary table 3]. For example, signal bleed through of a very bright channel into a dim channel will have a relatively big impact on the resulting image of the dim channel, but the other way around it will be hardly noticeable.

The model does not take into account the limited time window of the measurement, but assumes it to be infinite (deriving Eq. 5 from Eq. 3 and 4). Whereas for very short fluorescence lifetimes this assumption holds true, the longer the fluorescence lifetime turns, the less accurate the model will be. This leads to a loss in result precision for longer lifetimes. I suggest, similar to the table in Fig. 1a, to show a quantification of the precision to resolve between the FPs, with respect to the ratio of their fluorescence lifetime to the time window of the respective measurement.

We thank the reviewer for this observation. Indeed, for very long lifetimes one needs to acquire longer as the probability to recover photons with a very long delay between the excitation pulse and the detection is indeed very low. For all fluorescent proteins tested expressed in live cells, however, we had lifetimes no longer than 4.1 ns (Table 1-2). Perhaps for longer lifetimes one should consider the impact on the acquisition time and the frequency of the pulsed laser (for instance 80 Mhz – 12.5 ns might be too tight for longer lifetimes; in our case we employed a laser tuned at 20Mhz). We have now discussed these aspects and included the calculation of all methods: fitting, phasor and average lifetime [New Supplementary Figure 1 and 2]. For the FPs utilized throughout the paper we show that no major difference in the average lifetimes was obtained regardless of the method used.

Importantly, the excitation laser pulse repetition rate was set sufficiently low (20 Mhz) so that all fluorescence decays completely before the next laser pulse, as can be seen in the patterns (Supplementary Figure 4-7). Therefore, the accuracy of the results is the same for short and long lifetimes.

Related to this aspect: was the measurement time window kept constant for all experiments or was it adapted, for the different wavelengths employed for PIE or, eventually, depending on the fluorescence lifetime of the FPs used in the experiment?

The measurement window was kept constant for all experiments (200 frames, 3 minutes acquisition in total).

Both background and electronic noise of the detector impact on the accuracy of the result in FLIM evaluation. The model considers a background-free situation (Eq 1,2,3). Therefore, I suppose the background was subtracted from the measured signal, before processing the data using the non-fitting approach. How was the background value determined: in each image, at each timepoint after the laser pulse? As the background distribution should be a Poisson distribution (eventually a super-Poisson distribution), it is asymmetric. Hence, subtracting its mean value from the signal would affect in a different way fluorescence decays with shorter than with longer lifetimes. Therefore, it would be important to have a description of the steps performed prior the non-fitting model is applied. The instrument response function (IRF) should be shown, as the model considers it to be a delta-function. I expect the IRF to be narrow, however, its effect on shorter as compared to longer lifetimes should be demonstrated.

Thanks for these insightful remarks regarding both the noise and the accuracy of the shortest lifetimes and the instrument response function. Indeed, the noise was removed from the background signal employing the average number of counts after the pulse. With the HyD detectors employed the noise is very close to zero and on most occasions we did not need to remove it prior to process the average lifetime. We have also included our measured experimental IRF in our fitting calculations and the impact in measuring short lifetimes employing both fitting approaches, the phasor plot and the average lifetime approach [New Supplementary figure 1-2]. In our experience when measuring lifetimes between 1 and 4 ns the impact of the IRF is minimal as one can see when comparing the lifetimes in New Supplementary Figure 1.

Specific Comments:

Eq. 5 in the description of the algorithm for should read $\langle \tau \rangle = \frac{(1-fB) \cdot \tau_A^2 + fB \cdot \tau_B^2}{(1-fB) \cdot \tau_A + fB \cdot \tau_B}$, and not $\langle \tau \rangle = \frac{(1-fB) \cdot \tau_A^2 + fB \cdot \tau_B^2}{(1-fB) \cdot \tau_A + fB \cdot \tau_B}$. The final equation is however correctly derived, so that the results shouldn't be compromised.

Thanks for this remark, we have now included also the mathematical formulation for the multiple exponential approach and have a brand new section of material and methods with more equations, comprised the Phasor and Pattern Matching.

Reviewer #2 (Remarks to the Author):

Starling et al. describe a method to unmix simultaneous imaging of nine fluorescent proteins. They then use this to investigate HIV infection, first looking at the infectious synapse and then early replication.

The microscopy technique is out of my area of expertise and so I cannot comment on how well it works or what advantage it has over existing methods. However, I can say that it is quite difficult for a non-expert to follow the figures and understand what is being measured and calculated. A summarised simple explanation in the text would help.

We thank the reviewer for this useful comment. We have now modified all figures and included more information in each panel so that the non FLIM expert can understand and follow the logic for each result. We hope this will help readers from different backgrounds to follow the logic of the article in this new version.

Regarding the use of this technique to study HIV biology, in general it is hard to assess the

progression of infection as many of the controls are missing (for example, comparisons with unlabelled and individually labelled virions). I appreciate that the authors are trying to recapitulate published results, but I think they should control for their visualisation.

This is a very fair point. We have now produced and imaged different types of virus badges (single, double, and quadruple labelled). We have tested how the lifetime behaves in all these badges [New Figure 5]. Moreover, we have also employed the quadruple HIV-1 virions in new different imaging experiments to visualize HIV-1 infection steps [New Figure 6, 7 New Supplementary Figure 9].

Additionally, only one third of the particles were quadruple labelled, which might confound any conclusions someone using this technique could draw. This is not necessarily a fault with the imaging technique, but more a general issue with labelling viral particles.

We agree with the reviewer that perhaps the percentage of quadruple labelled particles is relatively small. We wanted to make a point regarding the evolution of the ones with four markers at different time points. Showing that some of them were able to fuse (losing the Gag-mCherry signal). Also, we managed to detect an increase of HIV-1 virions in the periphery of the nucleus with Vpr-mTQ2 and within the nucleus with CA-EGFP [New Figure 6 New Supplementary Figure 9] indicating that most likely these particles had been productive for fusion.

However, considering the controversy in the field over using the CA-GFP fusion (particularly over how much CA is labelled and what this fraction represents) and with whether the Vpr fusion localises with cores, there may be too much uncertainty with these labelling techniques (regardless of what label is used or how it is analysed) to make this really useful for understanding HIV biology at this time.

We agree with the reviewer regarding the whereabouts of Vpr and the controversy on CA-GFP labelling extent. We have now produced different badges with different labels **and** present new data comparing cells exposed to quadruple viruses at time zero with time = 90 min in three dimensions [New Figure 6 New Supplementary Figure 9]. Here, we can identify particles in the peripheral nuclear area (Vpr-mTurquoise2) and also a number of Capsid-EGFP within the nucleus. These particles are only detected 90 min after infection, confirming that the quadruple virus could be useful to dissect different entry steps. These results reproduce previous published data with CA-GFP as in [Burdick et al., 2020].

It is not our intention, however, to solve the controversy in the field regarding labelled Vpr-mTQ2 and CA-EGFP with this four labelled HIV-1 virus strategy. Therefore we just present the new results in terms of a semi-quantitative approach to dissect different HIV-1 entry steps. In that regard we look for three specific traits at different time points (0 and 90 min):

- Percentage of quadruple HIV-1 viruses
- Number of HIV-1 virus Gag-mCherry +
- Examples of CA-GFP within the perinuclear area

-Examples of Vpr-mTQ2 within the perinuclear area

We clearly state that for now, our intention with this labelling approach was to validate our new imaging approach rather to solve the meaning of the discrete viral products found at $t = 90$ min.

Not to mention the particle to infectivity ratio that implies most of the particles visualised are defective. Perhaps using an integrase label or following viral nucleic acid and watching particles colocalise with cellular markers would be more productive?

We agree with the reviewer that the majority of particles might be non-fusogenic. Indeed, we have experience in single virus tracking [Padilla-Parra et al., 2013, Jones et al., 2017 or Russel et al., 2017] and only a few percentage of the tracked particles are productive for fusion even when pseudotyped with highly promiscuous fusogens (such as VSV-G). Since the point of the paper was the validation of the multicolor approach employing a biological example we have not gone further and employed other markers in the context of HIV-1 postfusion events. We hope the new experiments will show clearly the validity of the quadruple labelled virus in the context of multiple color experiments [New Figures 6 and 7]. The new infectivity results confirm the reviewer's concern regarding infectivity, but in spite the drastic reduction in virus functionality we show the usefulness of this approach when locating sub-viral particles within the cell (New Figure 6).

Specific points:

Figure 3: The authors need to show singularly labelled cells expressing only CD4 or CXCR4 so that the reader can see what each signal looks like and then compare that to the digitally separated image in the co-expressed particles. It would help if the outline of the virus producing cell was indicated. They should also compare their synapse to a negative control ie with no synapse (using no env expression). Does the CD4 move? This really needs a comparison with a negative control or a time course in order to make sense.

We agree with the reviewer and present the proportion of CD4-mTFP1/CXCR4-mTQ2 in cells where the VS was not established. We present now these data in a new figure where the ratio of cells CD4-mTFP1 and CXCR-mTQ2 was quantified as a negative control [New Supplementary Figure 4]. Moreover, to strengthen our results, we have also quantified the proportion of receptors and co-receptors coming from outside the VS as a built in control (Figure 3e) employing three different FLIM analyses (fit, phasor and average). Both results clearly show that the abnormal proportion of CD4 and coreceptors observed within VS might be characteristic of the VS formation.

Figure 3: The figure legend wording is not clear and doesn't match the figure. Panels should be labelled A, B, C etc for clarity. The bar diagrams are confusing and the bottom left and

bottom right graphs are not even mentioned. What does “proportion” mean? How is this calculated?

We apologize for the confusion. We have followed the reviewers advise, included negative controls and also highlighted the infected cell for clarity. We have also corrected the figure legend and introduced the correct labels for each example. Every panel of the figure is now described and the different calculations explained.

The proportion is calculated now using three approaches: the average lifetime, the phasor plot and fitting procedures (previously validated in Figure 2). We hope that these calculations are easier to follow in this new version of the paper.

Figure 5a: The diagram does not show clearly where the DdD labels. The viral titre is meaningless unless there is a comparison with unlabelled virus. The authors should also show the titre of viruses singularly labelled with individual fusion proteins to see how each one affects viral infectivity. Are the images in this figure showing viruses on coverslips (i.e. not in infected cells)? This should be made clear

Thanks for these remarks. We have now modified the cartoon to stress where each label sits and their respective function the corresponding imaging experiment. We have produced different badges of viruses labelled with 1, 2, and/or 4 labels. We have performed new imaging experiments showing the validity of the quadruple labelled virus where we can assess the different steps of virus entry in a particular example [New Figures 5 -7].

We have also performed the corresponding infectivity assays evaluating each labelling approach and how the different markers affect infectivity relative to the wild type HIV-1 and also a negative control (No Env virus) [New Figure 5d].

We discuss these results in our new version of the manuscript. Nevertheless, we would like to stress that with the new imaging experiments [New Figure 6] we can pinpoint the particles that underwent fusion and appear to be within the nucleus only at late points after virus exposure ($t = 90$ min). We can image and identify these particles with high precision and hypothesize they were productive.

Figure 5b: The line profile doesn't appear to show quadruple labelled virus? The percentage co-localisation graph should show all the possible combinations of labels, ie how many particles were singularly labelled and each pairwise combination (even if some combinations had zero particles) to see the labelling efficiency. I would expect all particles to be Gag-mCherry and DiD positive if these are particles on coverslips? Therefore, I'm not sure that “double labelled” is the correct term for particles also containing CA-GFP?

We have now systematically produced different badges of HIV-1 viruses labelled with 1,2 and 4 labels and automated the analysis following the reviewer's advice. The new figure shows how the different badges are labelled, their colocalization and also their respective lifetimes.

Figure 5c: Do the images show particles produced with only two labelled constructs or just show the detection of two labels? This should be clarified.

Thanks for this comment. In the new figure we have tried to clarify the production and quantification of each label lifetime for each virus (single, double and quadruple labelled HIV viruses). We are convinced that in this new version of the figure one can follow the quantification, and there has been an improvement in the presentation of results. [New Figure 5].

Figure 6: I don't think we really know what a Vpr or CA signal on their own actually represent?

We can distinguish in the different micrographs discrete Vpr-mTurquoise signal that happens to be within cells and one can identify this due to the corresponding lifetimes characterized in the previous Figure 5 and new Figure 6. In the case of CA-GFP we have highlighted signal accumulating within the nucleus in our three-dimensional reconstruction coming from confocal microscopy [Supplementary Figure 9]. It is true that we cannot assign these signals a particular quadruple virus as these images were acquired 90 min after exposure. The images show however, that a sub-population of viruses was fusogenic and managed to reach the perinuclear area (as these signals were not seen at time zero). We agree with the reviewer that we should tone down the conclusions from these images regarding the importance of CA-GFP labelling, but clearly one can detect these viral proteins in subcellular structures and suggest different steps of entry.

Likewise, it is stated in the legend that particles with three colours (not DiD) represented fusion, but then shouldn't most of the mCherry also disappear? Can you measure the decrease in amount of mCherry staining? If it isn't lost, then it is even more unclear what the single labelled puncta are? As it stands, this analysis is not very useful as we cannot be sure what the staining means, especially as only a third of the particles start off with all four labels?

The reviewer is correct when stating that full fusion should be strictly related with mCherry-Gag disappearance. In fact we have calculated this and enhanced the statistics for this in the new experiments (New Figure 6). The global Gag-mCherry signal per particle indeed decreases suggesting that a fraction of the viral population undergoes fusion. We have performed these experiments in three dimensions and quantified all particles with high precision (New Supplementary Figure 9). Finally, we also show that in these particles two different lifetimes were calculated at $t = 0$ and the higher lifetime population diminishes at $t = 90$ min suggesting that the lifetime might be a good readout to predict fusion activity. We discuss these results in the new version of the manuscript.

Figure 6c: When it says "Single viruses for each label" does this mean particles made with just one label or detected on one channel? Please make this clearer throughout the paper.

We thank the referee for this comment. We have now corrected this throughout the paper and clarified the nomenclature of the labelled viruses. We have now fully characterized all

population of viruses. We hope this new analysis make our approach more clear [New Figures 5, 6 and 7].

Minor points

Line145: It states “the different percentages of colocalization were quantified in Figure 5a” - this is incorrect. I think it should be Figure 5b?

Thanks for this remark. With the new figures this has now been corrected.

Line 159: It would be helpful to state here what the lifetime of mTurquoise2 is in cells on its own.

Thanks for these remarks. With the new figures and explanations we hope to have solved these punctual doubts.

Line 205: I don't think virus producing cells normally have abnormal nuclei?

We have removed this sentence now.

Reviewer #3 (Remarks to the Author):

The manuscript of Starling et al demonstrates the power of combining lifetime and spectral information to multiplex the imaging of fluorescent proteins. Using their simplified approach, they have simultaneously measured nine different signals. They then apply their method to visualize the uptake and egress of HIV-1 virus like particles. Overall, the work has potential but some explanations and clarifications as needed before the manuscript can be considered for publication.

We thank the reviewer for potential strength of our new multicolor approach. We are convinced that thanks to the three reviewer suggestions we have clarified a number of issues and the new version of the manuscript is much robust and improved.

Major points:

1) The main deficiency in the current manuscript is the oversimplification of their approach and insufficient characterization of the method.

In the original manuscript the non-fitting method was compared with single and double fitting approaches for the blue fluorescent proteins (mTFP1 + mTurquoise2) in different experimental scenarios. Figure 2 and 3. We have now improved the comparison for the rest

of fluorescent couples (two species unmixing) using both the fitting approach and also the phasor plot as recommended by the first reviewer. We sincerely hope that the new comparisons will clarify some of the issues raised by the reviewer and will make the paper more robust.

As explained above, the non-fitting approach was previously benchmarked and developed for a two species model in which one sub-population was engaged in FRET. We have enlarged the previous model now for a two species model in which both species do not necessarily undergo FRET. We have cited the original paper [Padilla-Parra et al., 2008] and other papers where this approach was quantitatively compared with the Phasor Plot [Leray et al., 2013]. We have also improved the current paper introducing both the fitting analysis and the Phasor throughout the paper. Moreover, we also compare these approaches with the previously published Pattern Matching approach [New Figure 4]. Interestingly and thanks to the reviewer's suggestions we conclude now that both phasor plot analysis and the average lifetime approach are robust for simple models with two species [Figure 2 and 3] whilst the pattern matching approach (not employed in our previous version of the manuscript) turned out to be more robust for complex situations like imaging 9 or 7 fluorescent proteins simultaneously (New Figure 4 and 7). Interestingly, we have applied pattern matching analysis with a system not equipped with spectral detectors (see material and methods section).

The pattern matching approach has been successfully applied for multicolor imaging before with home-made systems equipped with spectral detectors as reported in:

Niehörster, T. et al. Multi-target spectrally resolved fluorescence lifetime imaging microscopy, *Nature Methods*, 257-262, 13(3), 2016

Rohilla. S. et al. Multi-target immunofluorescence by separation of antibody crosslabelling via spectral-FLIM-FRET, *Scientific Reports*, Vol.010, 3820, 2020

i) First of all, what is the relaxation constant, k_r ? Is this $1/\tau$? Is this a proportionality constant to convert decay events into collected photon intensity? Equation 1 is referring to "a single fluorophore", although I assume the authors mean a single type of fluorophore. It gets more confusing when the same k_r is used in Equation 2 for two types of fluorophores. Please clarify.

The reviewer is correct, k_r is $1/\tau$ and is valid for both single fluorophores or a mix of two species for examples with similar excitation emission properties. When considering the fluorescence decay over time, this global k_r would be experimentally recovered. In this scenario τ is what we call the average lifetime. If we have more than one species τ would contain the average of the two provided similar brightness and spectral properties. We have clarified this point now in the text and have cited seminal papers on the matter introducing the theory of the average lifetime for donors that behave as multiple exponentials [L820-L860].

ii) If you are describing the intensity, the brightness of fluorophore plays a role (determined by the probability of excitation, of detection and the quantum yield of the fluorophore). While the detection efficiency may be similar for dyes with similar spectra, the other terms will not be the same in general and the relative intensities will depend on their relative lifetimes. I understand the authors try to utilize fluorophores of similar brightness, but they should expand their calculations to correct for differences in brightness. This is straightforward and does not add to the complexity of the analysis when the information is already known. They should also discuss this in more detail.

We thank the reviewer for this suggestion. Indeed, as expressed in Figure 1b we have taken the brightness of each specific fluorophore as an important parameter to consider when unmixing two fluorescent proteins. The reviewer is also right when pointing out the importance of the intensity when recovering the signal for one or more fluorophores. Indeed, this is also important in lifetime imaging calculations. At least to recover a fluorescent decay with enough photons so that the average lifetime can be calculated properly.

When calculating the lifetime, however, and as long as the fluorescence decay is obtained with a good signal to noise and a high number of photons, the lifetime will be independent from the brightness of the fluorophore. This is so, because the lifetime is related to the decay, and not the height of the decay. In brief, when considering the log of the fluorescence decay we should be able to define the slope regardless of the number of photons. In practice, however, proteins spitting out several photons per excitation cycle will overtake the ones emitting less photons.

When evaluating the brightness of each FP, following the reviewer's suggestion, we found that the more important issue was to understand the crosstalk between channels rather than unmixing the different fluorophores per channel. This is why we had to introduce pattern matching, which was the only approach that took this aspect into consideration.

iii) Also, would it be possible to correct for FRET?

Yes, as explained above the original non-fitting approach was established for a two species system; one of them undergoing FRET [Padilla-Parra et al., 2008] and further compared with the phasor plot in [Leray et al., 2013]. We are now discussing this in more depth to clarify the foundation of the theory and its expansion to systems in which FRET does not necessarily happens.

Interestingly in a more complex situation (multicolor FLIM) FRET could also be incorporated into the pattern matching approach with reference patterns for donor and acceptor, for low and high FRET states.

iv) As the authors are using an analytical expression for calculating the ratio of lifetimes, they will always get a result. However, they authors should investigate how closely this represents the real situation and discuss the limits. (This will be discussed in more detail below). In FLIM, a factor 2 in lifetime is usually need to reliably separate two species. Of course, if you assume

two lifetimes, the algorithm will give you a result, but how significant is it (e.g. in distinguishing eGFP ($\tau = 2.4$) and mWasabi ($\tau = 3.1$ ns)). How does the reliability of their method depend up on the difference in fluorescence lifetime?

We thank the reviewer for this suggestion. We have now produced an experimental set up of different proportions of two species with different lifetimes to show how the system is capable of recovering the known proportions [New Supplementary Figure 3].

Regarding the accuracy of the model for species with very close lifetimes we have also taken this into account. In fact, the model should be able to separate two species with very close lifetimes, being the accuracy of the lifetime calculation the only limitation, as demonstrated in [New Supplementary Figure 3]. When measuring a particular lifetime there is an intrinsic error given by the system. We calculate this error employing the three approaches mentioned before (fitting approach, average lifetime calculation and the Phasor plot method) [New Supplementary Figure 1 and 2]. In all cases the limiting factor is the number of photons. Provided one has all photons needed the accuracy of the measurement for most fluorescent proteins expressed in live cells is around 100-200ps. This means that with our system lifetimes that overlap with this error will be very difficult to resolve.

In the case of the pattern matching approach, all reference patterns are used in the linear unmixing step. In case that a label is not present, the corresponding amplitude would be zero. In addition to the unmixed images, the algorithm shows the distribution of the residuals of the least squares pattern fit. These can indicate if the employed patterns were sufficient to describe the data in every image region. For example, low signal levels at the edge of cells or autofluorescence of certain structures could hinder the unmixing, leading to higher residuals. The pattern matching approach employs both lifetime information and spectral information, i.e., excitation wavelength and intensity per spectral detection channel. That makes it very reliable. Moreover, the pattern incorporates the full shape of the lifetime decay, which can be different even for fluorophores that have the same average lifetime but actually decay in a multi-exponential way.

v) The authors should also check the reliability and robustness of their approach to that of fitting each pixel to a biexponential decay (or to the phasor approach).

Thanks for this remark. We did this comparison in Figure 2(a and b). However this did not seem to come across as we (the authors) failed to communicate the results effectively in the original manuscript. We have now done this not only for the blue fluorescent proteins but also for the rest of two species model comprised in our PIE FLIM scheme [New Supplementary figure 1 and 2]. We have also included the Phasor Plot in all FLIM data presented [New Figures 2-7]. Finally, we have also compared our approach (average lifetime), the Phasor and fitting with the Pattern Matching approach (Niehurst et al., 2016, Nat Methods] for simultaneous 9 color imaging and 7 color imaging with viruses. We hope that this comprehensive comparative analysis will speak better regarding the robustness of our approach.

Importantly, all data was evaluated with multi-exponential fitting of each pixel in each detector channel (results not shown). The results were not as reliable and meaningful as those obtained with pattern matching.

2 Section: "Selecting FPs...". The authors discuss that having an exponential decay is important for their approach and they investigated 30 available FPs. It would be useful if the authors were present the exponential decays of the 30 FPs they investigated in the SI with mono-exponential fits and residuals so that the reader can get an idea of how mono-exponential the different FPs are.

We have now performed the fitting approach (single and double exponential model) for all FPs analysed in the paper. We have also included for the 9 chosen fluorescent proteins, the lifetime values for Average, Phasor and fit (New Figure 3d-f) and New Supplementary Figures 1 and 2.

3 Section: "Pixel by pixel unmixing...":

i) Here, there is no known ground truth. In addition, for the tandem, I would expect FRET. Have the authors checked for FRET.

By definition and given the overlap between mTurquoise2 emission and mTFP1 absorption spectra it is very likely there is FRET. However, it is very difficult to detect with FLIM as one needs to recover only photons coming from the donor. The 440 excitation laser excites both and therefore minimizes FRET to occur as the excitation state of the acceptor is already occupied. This scenario makes FRET almost impossible to quantify as the photons that are emitted by the acceptor (mTFP1) will compensate the ones emitted by the donor (mTurquoise2). Importantly, and in the case that FRET would be possible we should only see it in the mTFP1-linker-mTQ2 as the other two co-expressions both proteins diffuse independently. A proof that FRET is not happening is that with all methods (fitting, phasor and average lifetime) we recover similar proportions of each species (around 50% each) [Figure 2].

The set up was PIE with 440/485 lasers in which the 440 is able to excite both fluorophores and 485 only the mTFP1. Therefore, both species are easy to unmix. In fact when using both the fitting approach with two exponentials fitting one recovers for the three examples (tandem mTFP1-mTQ2, mTFP1-T2A-mTQ2 (in this case similar amounts of mTFP1 and mTQ2 should be expressed without a linker and therefore there is no FRET) and co-transfection) show around 50% for the three. We recovered a similar number with our non-Fitting approach (Figure 2b) and the Phasor (New Figure 2c). These aspects are now more thoroughly discussed in the new version of the manuscript (L130-160).

What are the brightnesses of mTFP1 and mTurquoise2 under these conditions? What excitation powers were used (this information should be given somewhere!). In addition, one does not know the maturation of the different proteins and transfection efficiencies when using different plasmids.

We thank the reviewer for opening this discussion. The brightness of mTFP1 is slightly higher as compared to mTurquoise2 (see Figure 1b). The key here is to understand that the mTFP1 was efficiently excited with 485nm laser whilst the mTurquoise2 with 440nm laser. We demonstrate that with this PIE strategy we can recover the population of each fluorescent protein independently. Indeed, the maturation time and other important factors related to spectral heterogeneity is not taken into account in the model, however we do see that in all cases we hit the 50% (+/- 3) mark which seems reasonable. Especially in the case of the bicistronic (T2A) and tandem plasmids where one would expect equal amounts of expression for both mTFP1 and mTQ2 as both proteins seem to have similar maturation rates [REF].

ii) To verify their methodology, the authors should artificially mix "pure" images to make a combined simulated image where the expected results are known and compare the results of their analysis. In addition, they should vary the contributions of the two species to explore the limits of their approach.

Thanks for this suggestion. We have now produced different systems with two species (varying both the proportion and the lifetimes) and have evaluated the error for two ideal situations testing also how well the average lifetime approach could be employed with two fluorophores with similar lifetimes (only separated 0.2 ns) [New Supplementary Figure 3]. Interestingly, we show that the most important parameter to control when applying our analytical approach is the signal to noise rather than a big increase in the lifetimes of the fluorophores evaluated.

4. Section: "Defining the proportions of HIV-1 receptors..."

Here are the first potentially interesting results. However, a lot of the useful information that could be extracted from such measurements is missing. i) What is the distribution of CD4 to CXCR4 expression in cells not exposed to viruses.

Thanks for the encouraging comments. We have now followed your (and reviewer 2) suggestion and present the results of the distribution of CD4/CXCR4 proportion before establishment of the virological synapse. [New Figure 3 and New Supplementary Figure 3]

ii) Does this correlate with the total concentration of CD4 or CXCR4? iii) The spatial distribution of viruses in Figure 3 is very inhomogeneous. Can the authors explain this?

In Figure 3 there is an infected cell that has engaged with the target cell generating the virological synapse where the proportion of CD4/CXCR4 is particular and different from the rest of the cell. This should have been better explained. We have now explained and discussed this in the new version of the manuscript.

We also apply the Phasor plot and fitting to these results and show that clearly the proportion of CD4 and CXCR4 is different right at the virological synapse (defined by Gag recruitment) as

compared to outside the virological synapse. (New Figure 3). We also include data of cells not engaged at all in the VS (New Supplementary Figure 4).

iv) In the presence of viruses, are their results significantly different than the 80%/20% you would expect? v) Does this ratio also vary with total concentration of the receptors available?

Thanks again for these insightful comments. Indeed, we stress the importance of the proportion of receptors and coreceptors in the VS as compared to the rest of the cell and also other negative controls (such as the proportion outside the VS, New Figure 3e). Indeed, this was already shown in the original figure, but we failed to explain this correctly. We have now modified the figure and added a little explanation about the matter so that these results are easier to follow. We also include three different FLIM analyses in the New Figure 3 obtaining similar results to validate our average lifetime approach.

In the past we have characterized the stoichiometry for CD4 and coreceptors with single viruses [Iliopoulou et al., 2018. NSMB]. We discuss the current results and compare them in the new version of the manuscript [L50-160].

5. Section: "Simultaneous Imaging of nine FPs..."

How do the authors know that what they measure is correct? Clearly the algorithm will give a result. Is there some other feature that one can be used to verify that the determined species is really the species of interest? (e.g. H2B will make dots in the nucleus, a plasma membrane marker will be clearly in the periphery, etc). In Figure 5, panel c, it appears as if there is a clear segregation of the different plasmids with only one plasmid present per cell. I would expect most cells to have incorporated all plasmids they have been exposed to, and most likely with similar ratios. Please explain why this is not the case.

Thanks again for these comments. As required by the reviewer it was necessary to compare across established methods with the one presented here based on the average lifetime. We have produced new figures and in this particular case [New Figure 4], we compare the Average Lifetime, the Phasor [Digman et al., 2008] and the Pattern Matching for FLIM [Niehurst et al., 2016]. In both the average lifetime and the phasor the degree of agreement is qualitatively seen when comparing the patterns for each individual fluorophore (for all two-species models) selected in each channel.

6. Section: "Single particle detection of quadruple labelled HIV-1..." Lines 197-199: "The second subpopulation of viruses engaged in the cells was further analyzed to unmix the viral and cellular compartments and detect different steps of the HIV-1 infection cycle." How is this done and where are these results?

In order to clarify a number of aspects required by Reviewer 2 we have produced a number of new figures in relation to the quadruple labelled virus application [New Figures 5-7]. We present results treated in three ways (Average lifetime, Phasor and Pattern Matching) to

compare the three approaches also in this context [New Figure 7]. We hope that the new explanations and comparisons will now be more clear and specific. We thank the three reviewers to bring this issue to our attention.

7. Figure 7 does not contribute new information to the work and can be removed. This is a summary of things already known.

We have now removed old Figure 7 following the reviewers advice.

8. There are many relevant studies that have combined FLIM and spectral imaging, or used fluorescence lifetime to investigate viruses that should be cited within this paper. The citations are very egocentric, focusing on work from the Padilla-Parra lab and not on the general literature.

We apologize for this. The excess of self-esteem in our own work could have been misunderstood as egocentric or as a lack of humbleness. It was not done in purpose. It so happens that most of this work presented is rooted in our previous published research naturally. Of course, there is no excuse to neglect other important contributions in the literature taking into consideration other multicolor FLIM approaches. We have now referenced more articles and only call our papers in the text when strictly required. We hope this version of the manuscript is more balanced in that regard.

Minor points:

1. Lines 43 and 260: The authors twice mention artificial intelligence, but other than throwing out the name, there is not substantiation of how they expect AI to contribute. This should either be more appropriately discussed with proper references in the discussion/outlook or removed from the paper.

We thank the reviewer for this remark. We have now cited a couple of recent papers on the matter [Heliot and Leray 2019, Xiao et al., 2023] and include a brief discussion [REF] [L596-599].

2. Lines 49-51: As the authors know, the necessity of a single exponential decay is not necessary when analyzing with the phasor approach. The authors should mention this in the introduction.

We have now compared the average lifetime, the phasor plot and the fitting approaches and shown the strengths and weaknesses of all three methods in different scenarios. Indeed, the phasor plot does not necessitate single exponential approaches as does not the average lifetime either. Actually, the phasor plot is a way of applying the average lifetime in two dimensions, phase and modulation (frequency domain) FLIM methods as discussed above.

In fact, for a fluorescent protein to behave as a single exponential is still a good thing regardless the analysis method employed. This is so, because a single exponential behavior

(which would imply that the fluorophore fall into the universal curve of the Phasor) is an intrinsic characteristic that denotes a positive feature for imaging, since there is only one species being excited. This instance was studied and discussed in [Tramier et al., 2005] for the CFP as a paradigmatic example of multiexponential FP. In this particular case, photobleaching would affect more one species than the other (as CFP is known to behave as a double exponential) therefore a change in lifetime (that by the way would also affect S and G in the phasor) will happen regardless other phenomena such as FRET or changes in the refractive index. And this is what we mean in this particular case. This paper from Tramier and colleagues was cited in the introduction.

3. Line 208: "Nucleoplasmic"

Thanks for this, we have corrected this typo.

4. Line 394, Equation 6: tau should be $\langle \tau \rangle$ in the formula.

This has been corrected

5. Figure 1: Panel a, right side. I do not understand these tables and what is being shown. e.g. the upper left box for blue excitation has a lifetime of 1.9 ns and a label of mTFP1 on top and mTQ2 on the left. What does this lifetime refer to?

We apologize for this. These lifetimes show the increase in lifetimes between two fluorophores to be resolved (in this case mTFP1 – mTQ2). When the increase is bigger than 0.4 ns we consider this difference to be good enough to separate two fluorophores based on their respective lifetimes and we assign a green color. When the lifetimes increase gets closer and closer we denote these colors orange (0.2-0.4 ns) and red (less than 0.2ns) respectively. We have included an explanation in the Figure legend now for clarity.

6. Figure 2: i) Using a) b) and c) for panels and to simultaneously to discuss parts of a single panel is confusing. Please use a different notation within a panel (e.g. i, ii, iii...). ii) What is being shown in the middle sub-panels of panel a. Information/labels and description are missing here.

We thank the reviewer for this remark. We have now modified both the figure and the figure legend to clarify this point.

7. Figure 5, Panel a, right side: The X-gel assay should be better described. I do not understand what is being shown in this figure (I can only guess).

Thanks for these fair remarks. This figure has completely removed and we have now presented the infection data comparing the infectivity with a positive and negative controls together with different labelling strategies. We hope this new figure is easier to read and follow now.

8. Figure 6, Panel a, right side: The arrows are confusing. Better would be data points. Please

also plot the number of viruses for viruses on the coverslip (so one can see that they remain constant).

Thanks again for these thorough comments. Figure 6 and 7 are now brand new. The calculations for the number of particles and the lifetimes are now easier to follow and hope this will answer this fair point.

9. Figure 6, Panel b, fourth row: There are cells visible in the FLIM image that are not described as subpopulations. I assume this is cross talk of mScarlet into the red channel. However, the authors need to explain how they deal with this and not just ignore it. As often in the paper, important details are missing.

We have now produced a new figure analysed with both the Phasor Plot and the average lifetime. In this case, we show how each channel is treated and separated in all cases. We hope that the newer version of the manuscript is now more accurate regarding FLIM calculations.

10. Line 604: Supplementary Figure 4: "and their errors increase from 200 photons per pixel". I have never heard before of errors in a fit increasing when the number of photons increases. Please reword.

Thanks for this remark. We have now removed this sentence since as pointed by the referee it was confusing.

REVIEWERS' COMMENTS:

Reviewer #1 (Remarks to the Author)

In their revised manuscript on "Visualizing HIV-1 entry with multicolor lifetime imaging" the authors fully addressed my concerns regarding the reliability and performance of their multicolor FLIM evaluation approach and, thus, substantially improved the manuscript, making it in my opinion adequate for publication.

Reviewer #2 (Remarks to the Author)

In general, although improvements have been made to the manuscript, I still have issues with the biological relevance of this study. As the infectivity of quadruple-labelled particles is much lower than the unlabelled virus (what is the y-axis in Fig 5D, the numbers are strange?), then I'm not really sure what these particles represent, or therefore whether their subcellular location means anything, and I doubt in reality this would be useful for studying viral replication as there are so many assumptions. This might be a better technique for looking at viral synapses, but although the authors mention hemifusion etc, they don't actually look at individual stages of fusion and I'm not sure they could resolve these? Although they have changed the title, it is still misleading, as they don't really visualise the entry process. In summary, I think this this would be better to report as a methods paper really.

Some additional minor points:

The authors should be careful about the wording of labels - when they use target cell, infected cell, effector cell etc.

Although Vpr has been reported to do many things, I don't think stabilizing the capsid is one of them! This sentence should be removed (line 305).

Some of the new passages have very poor English grammar (especially lines 357-364) that needs addressing to make sense.

Reviewer #3 (Remarks to the Author)

Through the input of the reviewers, the paper has been much improved. However, there are a number of issues that still need to be improved.

Major Issues

1. For the main FPs used in this study, it would be useful to have the fluorescence excitation and emission spectra in the SI.
2. I had asked the authors to correct for the fluorescence brightness in their calculations. When separating multiple species within a single channel and pixel, the fits determine the photon contributions to the fluorescence decay. To convert this into a population, the brightness needs to be taken into account. The authors should discuss this point and discuss how the conversion is made for the different methods.
3. L111-115: " Even if Forster Resonance Energy Transfer (FRET) could occur between mTFP1 and mTurquoise2, this phenomenon could not be quantified as both the donor and the acceptor are excited with the 440 nm pulsed laser, minimizing the options of transfer as the acceptor excited state (in this case the mTFP1) is already populated with electrons by both excitation lasers (440 and 485nm)." This statement is incorrect and should be reworded. If indeed the acceptor is always excited at the same time as the donor, this would imply that the authors are measuring under saturation conditions and a quantitative analysis of the data is impossible. As I am sure the authors are smart enough to not measure under saturating conditions, hence, FRET is possible and

should be occurring for a tandem with a separation of 16 aa (see, e.g. Coullomb et al, Scientific Reports 2020). It is very possible that FRET occurs between mTFP1 and mTurquoise2 in both directions with similar probability, at least with 440 nm excitation, and as such, is not detectable to them. (To estimate this, having the spectra in the SI would have been helpful). To determine whether FRET is occurring under these conditions, anisotropy measurements would be needed as in homoFRET. An additional measurement is not requested here (unless the authors want to satisfy their own curiosity), but they should rewrite the text. A tandem separated by 16 aa will undergo FRET.

4. L229-231: "This kind of evaluation of the unmixing performance is not possible for the average lifetime or the phasor plot approach, since it yields no amplitude values, only information whether a FP is present in an image pixel or not." This statement is incorrect. The phasor plot allows you to determine the contribution from multiple components. It works extremely well for two species, but may become more challenging when more species are present. Please reword.

5. L251-252: "Surprisingly, the single labeled virus (HIV-1 (Vpr-mTQ2) presented higher infectivity as compared to the wild type." Where is the evidence for this? I do not see a titer plot. I would also recommend that the authors remeasure this as it cannot be that the labeled virus is more infectious than the wild type. This statement endangers the authors' credibility to the virology community.

6. L256-258: "We speculate that the viral environment must have a big impact in mTurquoise2 and Capsid-GFP since the lifetime of mTQ2 when expressed alone was 4.1 ns." It has been shown that attachment FPs to viral proteins already influences the lifetime and that the HIV environment can influence it even more, at least for some FPs. Although the authors are aware of this work supporting their speculation, they chose not to mention it or cite it. In general, there are still relevant papers in the field that are not cited by the authors in the revised manuscript.

7. The authors should include zoom ins of individual cells to give the authors a better resolution of the observed sub-cellular structures. One of the most convincing arguments that the authors can indeed resolve multiple species in a single image is the different structures that become observable after separation of the image into its components (e.g. into microtubules and mitochondria structures). Currently, the microtubule and mitochondrial staining are not convincing, at least at the resolution plotted in Figures 4 and 7.

Minor Issues

1. The authors mention in several places "one-shot" imaging of multiple colors. It would be more accurate to say that multiple species can be followed in a single experiment. These measurements are not "one-shot" implying that no other measurements are needed. For pattern recognition etc, the individual information for each fluorophore is needed. In addition, as nicely mentioned in L256-258, labeling and the viral environment can influence the lifetime. Hence, the fluorescence lifetime needs to be calibrated on the system of interest. Therefore, multiple species can be measured in a single experiment, but not in a "single shot". Please reword.

2. L82-84: "From a panel of 30 available FPs (Supplementary Table 1) were individually expressed and imaged by FLIM in live cells." - Sentence is grammatically incorrect.

3. L150: "from"

4. L642, Eqn 3: Subscripts are messed up for α and τ . Also in the text on line L646.

5. L650, Eqn 4: A bracket is missing in the numerator.

6. L669, Eqn 1: Equation numbering is degenerative.

7. L676, Eqn 2 and L684, Eqn 3: Problems again with subscripts.

8. L702, Eqn 5, L707, Eqn 6: Again, problems with subscripts.

9. Note: I have not checked the equations in detail and, considering the obvious mistakes that have escaped the authors' attention, I would recommend they check all equations carefully.

10. Figure 1a: "The right tables.." What is plotted on the right is a diagram, not a table. Also, the authors should add the lifetime of the individual FPs with the labels of the FPs (e.g. in a second line above/beside the squares).

11. Figure 11b: "We employed two fluorescent proteins per channel with the right spectral, lifetime and brightness (b) properties to achieve more than eight multicolor imaging in one." Can the authors expand on this. From three detection panels alone, they would have six species, not eight. They are also including PIE channels as well.

12. Figure S1: Please clearly identify the constructs being measured here, not only the FP. In addition, it is also not clear what is being plotted in the right most histograms of panel a. (The 1.500 label is also confusing. Why not 1.5?)

13. Figure S2a: Incorrect title: "Atto647N + Atto488"

14. Figure S2: Plots of fraction versus solution. The plots are confusing. Please plot the data as the theoretical fraction (x) versus experimental fraction (y) along with a line of slope 1 to guide the eye.

RESPONSE TO REVIEWERS' COMMENTS

Reviewer #1 (Remarks to the Author)

In their revised manuscript on "Visualizing HIV-1 entry with multicolor lifetime imaging" the authors fully addressed my concerns regarding the reliability and performance of their multicolor FLIM evaluation approach and, thus, substantially improved the manuscript, making it in my opinion adequate for publication.

Reviewer #2 (Remarks to the Author)

In general, although improvements have been made to the manuscript, I still have issues with the biological relevance of this study. As the infectivity of quadruple-labelled particles is much lower than the unlabelled virus (what is the y-axis in Fig 5D, the numbers are strange?), then I'm not really sure what these particles represent, or therefore whether their subcellular location means anything, and I doubt in reality this would be useful for studying viral replication as there are so many assumptions.

We thank the reviewer for acknowledging the improvements made in the manuscript. We have put a lot of effort in trying to answer all questions from the previous round with new experiments and data that in our opinion clearly shows the validity of our approach combining multicolor FLIM and a quadruple-labelled virus. We have produced new infectivity data comparing the labelled virus with the non-labelled wild type (Empty) (**New Figure 5d**). Clearly the quadruple labelled virus still is quite infectious. Regardless a big drop in infectivity, the titers for the quadruple-labelled virus shown in the infectivity assay (**New Figure 5d**) are relatively high (bigger than 10^6 infectious units/ml).

We have clearly shown that the use of the quadruple-labelled virus is very useful for three main reasons: 1) Precise localization of the virus when exposed to the cell at different time points is given (as shown in **Figure 6a**). 2). We can detect post-fusion events in three dimensions. Sub-viral particles such as Vpr-mTQ2 and CA-eGFP were only detected around and within the nucleus respectively at late time points after virus exposure ($t = 90$ min). (**Supplementary Figure 9**). 3) We show that we can quantify Gag-mCherry disappearance which in turn is related to Gag-mCherry lifetime shortening and is a proxy for productive fusion (**Figure 6b-c**) (as emphasized by the reviewer in the previous round of comments and also employed by us and others in the literature [Jones et al., 2017, Russell et al., 2017; Padilla-Parra et al., 2012, Miyauchi et al., 2009]). These data, therefore, strongly suggests that quadruple-labelled virus can be employed to measure productive entry.

This might be a better technique for looking at viral synapses, but although the authors mention hemifusion etc, they don't actually look at individual stages of fusion and I'm not sure they could resolve these?

Thanks for this suggestion. We have produced a figure that shows that the quadruple-labelled particles are capable to undergo hemifusion and full fusion when pseudotyped with both JRFL Env and VSV-G. (Fig. 1 rebuttal, below)

Fig. 1 rebuttal. Real time single virus tracking with multilabelled HIV-1 particles decorated with JRFL and VSV-G. (A) Cartoon showing how the different labels change in color when hemifusion (white to light blue), full fusion (light blue to dark blue) or endosomal fusion (color separation red and dark blue). (B) In the upper panels HIV-1 quadruple labelled (as in Figure 5) was exposed to live T cells. A single virus was tracked (white arrow) that undergoes hemifusion (third panel from the left, 0) and full fusion (fourth panel from the left, 1). The bottom panels show HIV-1 particles decorated with VSV-G (known to fuse within endosomal particles). The white arrow clearly shows a particle undergoing endosomal fusion where color separation occurs (capsid release, 2). Scale bar 1 μ m.

Clearly, a fraction of the quadruple labelled particles was able to undergo both hemifusion and fusion (as explained above) and demonstrated in the previous figure. We feel however, that real-time single virus tracking experiments, solely based on intensity, with these particles are out of the scope of the present work as we wanted to put emphasis in the multicolor FLIM applied in different virological scenarios (e.g. virological synapse and localisation of sub-viral particles during entry). Therefore, we have left out these experiments from the current manuscript, as they are out of the scope of the paper and deviate the attention of the focus of the paper. Real time single virus tracking combined with multicolor FLIM is ongoing research in our lab that still needs more events to be validated. Fig. 1 rebuttal, however, shows that a sub-population of these quadruple-labelled particles undergoes hemifusion and full fusion. (Note that dark blue contains both Vpr-mTQ2 and CA-eGFP in our color code).

Although they have changed the title, it is still misleading, as they don't really visualise the entry process. In summary, I think this this would be better to report as a methods paper really.

We appreciate the reviewer's opinion; however, we are convinced that the two biological examples provided (the **stoichiometry of CD4** when engaged in the **virological synapse** and identification with **multicolor lifetime of different stages during virus entry**, which comprise post fusion events) are very important and timely within the HIV-1 community.

Some additional minor points:

The authors should be careful about the wording of labels - when they use target cell, infected cell, effector cell etc.

Thanks for this suggestion. We have now employed target cell and infected cell exclusively throughout the manuscript.

Although Vpr has been reported to do many things, I don't think stabilizing the capsid is one of them! This sentence should be removed (line 305).

We agree with the reviewer and therefore have removed this sentence from the manuscript.

Some of the new passages have very poor English grammar (especially lines 357-364) that needs addressing to make sense.

Thanks for this insightful remark. We have scrutinized this paragraph and others from the manuscript and hope the grammar structure is now richer after a few corrections.

Reviewer #3 (Remarks to the Author)

Through the input of the reviewers, the paper has been much improved. However, there are a number of issues that still need to be improved.

We thank the reviewer for the encouraging words and for taking the time to thoroughly read the new version of the manuscript. We have also taken seriously the new comments and tried to address all of them in full.

Major Issues

1. For the main FPs used in this study, it would be useful to have the fluorescence excitation and emission spectra in the SI.

This is a nice suggestion, and we agree that these spectra will be useful for the reader. We have therefore included those in the new **Supplementary Figure 12**

2. I had asked the authors to correct for the fluorescence brightness in their calculations. When separating multiple species within a single channel and pixel, the fits determine the photon contributions to the fluorescence decay. To convert this into a population, the brightness needs to be taken into account. The authors should discuss this point and discuss how the conversion is made for the different methods.

We thank the reviewer for these suggestions and apologize not to have explained this better in the previous version of the manuscript. As explained in the discussion and in (Niehorster et al., 2016) [L377 – L386] pattern analysis takes into consideration excitation laser wavelength and spectral detection channel together with lifetime information of each FP under consideration. This information is taken into account in a reliable single-step procedure to deliver quantitative results showing both localizations and relative intensity contributions for all FPs. Here the method for intensity correction is built-in in the algorithm. The novelty here is the combination of PIE and the right choice of FP's to employ a commercial system equipped with three detectors as a spectral system capable of imaging up to 9 FPs simultaneously.

To follow the reviewer's suggestion, we have included a new section in material and methods [**Calculation of photon contribution of each fluorophore species**]. In this section we introduce the way we have obtained the percentage of each species for each channel for fitting procedures and the average lifetime approach. We also include the brightness correction as suggested by the reviewer. This could also be applied to the phasor provided the user has access to a pixel-by-pixel image with the fraction contribution for each species.

3. L111-115: " Even if Forster Resonance Energy Transfer (FRET) could occur between mTFP1 and mTurquoise2, this phenomenon could not be quantified as both the donor and the acceptor are excited with the 440 nm pulsed laser, minimizing the options of transfer as the acceptor excited

state (in this case the mTFP1) is already populated with electrons by both excitation lasers (440 and 485nm)." This statement is incorrect and should be reworded. If indeed the acceptor is always excited at the same time as the donor, this would imply that the authors are measuring under saturation conditions and a quantitative analysis of the data is impossible. As I am sure the authors are smart enough to not measure under saturating conditions, hence, FRET is possible and should be occurring for a tandem with a separation of 16 aa (see, e.g. Coullomb et al, Scientific Reports 2020). It is very possible that FRET occurs between mTFP1 and mTurquoise2 in both directions with similar probability, at least with 440 nm excitation, and as such, is not detectable to them. (To estimate this, having the spectra in the SI would have been helpful). To determine whether FRET is occurring under these conditions, anisotropy measurements would be needed as in homoFRET. An additional measurement is not requested here (unless the authors want to satisfy their own curiosity), but they should rewrite the text. A tandem separated by 16 aa will undergo FRET.

We thank the reviewer for this remark about FRET between mTFP1 and mTQ2. We agree that most likely there is FRET between both mTFP1 and mTQ2. We were, perhaps, not precise enough in our disquisitions regarding the reasons for the absence of detectable FRET in our system. We have amended the text following the reviewer suggestion and now we allude only to the fact that we could not detect photons coming from the donor/s alone for this particular couple given the emission filters utilized. Therefore FLIM might be not the best technique to quantify FRET for this particular FP couple as blue photons will always be captured in the donor channel. We will explore homoFRET with these tandems in the future, but feel is out of the scope of this work.

4. L229-231: "This kind of evaluation of the unmixing performance is not possible for the average lifetime or the phasor plot approach, since it yields no amplitude values, only information whether a FP is present in an image pixel or not." This statement is incorrect. The phasor plot allows you to determine the contribution from multiple components. It works extremely well for two species, but may become more challenging when more species are present. Please reword.

We apologize for this erroneous sentence. Indeed, both the phasor and the average lifetime approach are able to unmix different populations of fluorophores with high accuracy for two species (as shown across the paper). We have removed this sentence from the paper.

5. L251-252: "Surprisingly, the single labeled virus (HIV-1 (Vpr-mTQ2) presented higher infectivity as compared to the wild type." Where is the evidence for this? I do not see a titer plot. I would also recommend that the authors remeasure this as it cannot be that the labeled virus is more infectious than the wild type. This statement endangers the authors credibility to the virology community.

When performing this assay, we produced the different labelled viruses on one hand and compared this with another badge (unlabelled "Empty") that was produced previously with different transfection conditions. Clearly, the number of physical particles for the unlabelled badge was different (lower) and therefore the infection percentage was indeed inferior. We have now amended this error with new X-Gal experiments that clearly show the unlabelled virus presents a higher titer of infectivity as compared to the labelled ones (**New Figure 5d**).

6. L256-258: "We speculate that the viral environment must have a big impact in mTurquoise2 and Capsid-GFP since the lifetime of mTQ2 when expressed alone was 4.1 ns." It has been shown that attachment FPs to viral proteins already influences the lifetime and that the HIV environment can influence it even more, at least for some FPs. Although the authors are aware of this work supporting their speculation, they chose not to mention it or cite it. In general, there are still relevant papers in the field that are not cited by the authors in the revised manuscript.

We have now cited the paper mentioned by the reviewer (Qian et al., 2022). Indeed, we are aware of the excellent work of the authors and we missed citing this important contribution in regards the impact of Gag labelling on lifetime when incorporated in HIV-1 individual particles. Currently, we have cited 39 important reports related to the work and we feel that these citations are balanced. However, we are sure we are missing the work of some important groups and apologize in advance for it.

7. The authors should include zoom ins of individual cells to give the authors a better resolution of the observed sub-cellular structures. One of the most convincing arguments that the authors can indeed resolve multiple species in a single image is the different structures that become observable after separation of the image into its components (e.g. into microtubules and mitochondria structures). Currently, the microtubule and mitochondrial staining are not convincing, at least at the resolution plotted in Figures 4 and 7.

Thanks for this suggestion. We have produced a new figure with blow ups from Figure 4 that clearly show the organelles stained with the different markers utilized (Histone H4, Histone H2B, Lamin, Tubulin and mitochondria). These labels were mixed with other fluorescent proteins without particular localization that have not been highlighted. We are convinced that the makers highlighted in **New Supplementary Figure 11** are easy to recognize. Note, however, that we were employing HEK293T cells to maximize expression of FPs employing transient transfection protocols, however we were limited by the cells shape and size (small and round).

Minor

Issues

1. The authors mention in several places "one-shot" imaging of multiple colors. It would be more accurate to say that multiple species can be followed in a single experiment. These measurements are not "one-shot" implying that no other measurements are needed. For pattern recognition etc, the individual information for each fluorophore is needed. In addition, as nicely mentioned in L256-258, labeling and the viral environment can influence the lifetime. Hence, the fluorescence lifetime needs to be calibrated on the system of interest. Therefore, multiple species can be measured in a single experiment, but not in a "single shot". Please reword.

We thank the reviewer for this remark. We agree with these suggestions, and we have amended the text accordingly.

2. L82-84: "From a panel of 30 available FPs (Supplementary Table 1) were individually expressed and imaged by FLIM in live cells." - Sentence is grammatically incorrect.

Thanks! We have corrected this sentence.

3. L150: "from"

Thanks for spotting this typo. We have corrected it now.

4. L642, Eqn 3: Subscripts are messed up for ai and tau. Also in the text on line L646.

5. L650, Eqn 4: A bracket is missing in the numerator.

6. L669, Eqn 1: Equation numbering is degenerative.

7. L676, Eqn 2 and L684, Eqn 3: Problems again with subscripts.

8. L702, Eqn 5, L707, Eqn 6: Again, problems with subscripts.

Given the fact that we put together these equations with word, we have experienced some problems with the editor for equations. Originally, we developed and wrote all these equations with other text editors (LateX). We have now been extra-careful with subscripts and numbering to address these typos. Thanks for spotting those!

9. Note: I have not checked the equations in detail and, considering the obvious mistakes that have escaped the authors' attention, I would recommend they check all equations carefully.

Thanks again for your advice! We have checked the equations and indeed only problems with subscripts have been identified and corrected.

10. Figure 1a: "The right tables.." What is plotted on the right is a diagram, not a table. Also, the authors should add the lifetime of the individual FPs with the labels of the FPs (e.g. in a second line above/beside the squares).

Thanks for this suggestion. We have now modified the figure to follow the reviewer's advice (**New Figure 1a**)

11. Figure 11b: "We employed two fluorescent proteins per channel with the right spectral, lifetime and brightness (b) properties to achieve more than eight multicolor imaging in one." Can the authors expand on this. From three detection panels alone, they would have six species, not eight. They are also including PIE channels as well.

Indeed, we were employing PIE channels and large stokes shift proteins (LSSmKate2 and LSSmOrange). This was explained originally in Results:

"By using large Stokes shift FPs (LSS-FPs) we were able to employ the red detector twice (one for the 440 nm line and another for the 594 nm) and increase the number of FPs to be multiplexed."

Basically, the combination of PIE and LSS fluorescent proteins allowed us to increase the number of proteins to be imaged with the red detector (one couple for laser 440 nm (LSSmOrange + LSSmKate2) and one for laser 594 (mKate2 + mCherry). Like this we have 8 proteins. But we also have Ametrine which is another LSS FP that we employ in the same way with the green detector:

440 ex (green emission) Ametrine / 488 excitation (green emission GFP and Wasabi). Like this we have 8 + 1 colors coming from genetically encoded FPs.

We have clarified this aspect now in the text [L197 – L200].

12. Figure S1: Please clearly identify the constructs being measure here, not only the FP. In addition, it is also not clear what is being plotted in the right most histograms of panel a. (The 1.500 label is also confusing. Why not 1.5?)

The only construct that is not a FP alone is mitochondria-LSSmKate2. We have now identified this in the figure legend. The histograms are the lifetime histograms coming from the FLIM images and the bar diagrams the average lifetime coming from 5 independent acquisitions. This is explained in the corresponding figure legend.

13. Figure S2a: Incorrect title: "Atto647N + Atto488"

14. Figure S2: Plots of fraction versus solution. The plots are confusing. Please plot the data as the theoretical fraction (x) versus experimental fraction (y) along with a line of slope 1 to guide the eye.

Thanks again for your thorough revision and suggestions to improve the paper. We have performed these changes as requested (**New Supplementary Figure 3**).

REVIEWERS' COMMENTS

Reviewer #2 (Remarks to the Author):

Following the latest round of revisions, figure 5 is much improved and now the scale has been fixed for the graph showing the titres of the viruses so it can be understood. However, I feel the authors still don't get the point that their viruses might not be representative of natural infections. The authors claim in their rebuttal that the titres are still relatively high (over 10^6 infectious units per ml) but they are being somewhat disingenuous as the viruses have been concentrated 100 times before titring (Unless you scrutinise the methods, you wouldn't realise this). Therefore, the depicted titre of 10^6 is in fact 10^4 . Regardless of the actual number, more important is the relative comparison to unlabelled virus. Unlabelled virus is over 100 times more infectious, clearly adding labels is inhibiting the virus. In most viral research labs, a 100-fold drop in titre would be considered non-infectious! Indeed, other studies that label virus particles go to great lengths to make sure their viruses have at most only 2-3 fold lower titres. The biological relevance of what the authors observe is therefore called in to question. This is not a reflection on their method of simultaneously monitoring several proteins, but more a general problem of labelling viruses. This is why I previously said that the paper would be better as a methods paper. At the very least, the authors should acknowledge this problem properly.

Reviewer #3 (Remarks to the Author):

The authors have done a good job with implementing the recommendation of the reviewers. I agree with Reviewer 2 that the focus of the paper is more on the methodology than on the virology. Hence, a title such as "Multicolor Lifetime Imaging and its Application to HIV-1 Uptake" may be more appropriate. Where caution may be necessary regarding the virological relevance of the study, it is a beautiful demonstration of the type of experiments that become possible with multicolor imaging.